# NeuralFuse: Learning to Recover the Accuracy of Access-Limited Neural Network Inference in Low-Voltage Regimes

## Abstract

Deep neural networks (DNNs) have become ubiquitous in machine learning, but their energy consumption remains a notable issue. Lowering the supply voltage is an effective strategy for reducing energy consumption. However, aggressively scaling down the supply voltage can lead to accuracy degradation due to random bit flips in static random access memory (SRAM) where model parameters are stored. To address this challenge, we introduce **NeuralFuse**, a novel add-on module that addresses the accuracy-energy tradeoff in low-voltage regimes by learning input transformations to generate error-resistant data representations. NeuralFuse protects DNN accuracy in both nominal and low-voltage scenarios. Moreover, NeuralFuse is easy to implement and can be readily applied to DNNs with limited access, such as non-configurable hardware or remote access to cloud-based APIs. Experimental results demonstrate that, at a 1% bit error rate, NeuralFuse can reduce SRAM memory access energy by up to 24% while recovering accuracy by up to 57%. To the best of our knowledge, this is the first model-agnostic approach (i.e., no model retraining) to address low-voltage-induced bit errors.

## 1 Introduction

Energy-efficient computing is a primary consideration for the deployment of Deep Neural Networks (DNNs), particularly on edge devices and on-chip AI systems. Increasing energy efficiency and lowering the carbon footprint of DNN computation involves iterative efforts from both chip designers and algorithm developers. Processors with specialized hardware accelerators for AI computing are now ubiquitous, capable of providing orders-of-magnitude more performance and energy efficiency for AI computation. In addition to reduced precision/quantization and architectural optimizations, low voltage operation is a powerful knob that impacts power consumption. There is ample evidence in computer engineering literature that study the effects of undervolting and low-voltage operation on accelerator memories that store weights and activations during computation. Aggressively scaling down the SRAM (Static Random Access Memory) supply voltage below the rated value leads to an exponential increase in bit failures, but saves power on account of the quadratic dependence of dynamic power on voltage (Chandramoorthy et al., 2019; Ganapathy et al., 2017). Such memory bit flips in the stored weight and activation values can cause catastrophic accuracy loss.

A recent spate of works advocates low voltage operation of DNN accelerators using numerous techniques to preserve accuracy ranging from hardware-based error mitigation techniques (Chandramoorthy et al., 2019; Reagen et al., 2016) to error-aware robust training of DNN models (Kim et al., 2018; Koppula et al., 2019; Stutz et al., 2021). On-chip error mitigation methods have significant performance and power overheads. On the other hand, some have also proposed to generate models that are more robust to bit errors via a specific learning algorithm (Kim et al., 2018; Koppula et al., 2019; Stutz et al., 2021), thereby eliminating the need for on-chip error mitigation. However, error-aware training to find the optimal set of robust parameters for each model is time and energy-intensive and may not be possible in all access-limited settings.

In this paper, we propose a novel model-agnostic approach: *NeuralFuse*. NeuralFuse is a machine learning module that allows for mitigating bit errors caused by very low voltage operation, through a trainable input transformation parameterized by a relatively small DNN, to enhance the robustness of

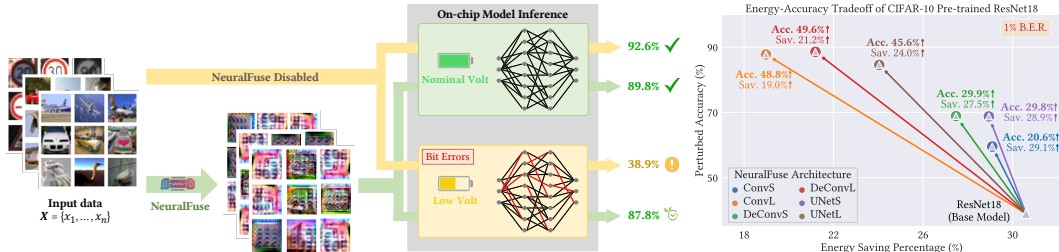

(a) The pipeline of the NeuralFuse framework at inference.     (b) Energy-accuracy tradeoff example.

Figure 1: (a) At inference, NerualFuse transforms input samples **x** into robust data representations. The *nominal* voltage allows models to work as expected; however, the *low-voltage* one would encounter bit errors (e.g., $1\%$) and cause incorrect inference. The percentage illustrates the accuracy of a CIFAR-10 pre-trained ResNet18 with and without equipping NeuralFuse in both cases. (b) On the same base model (ResNet18), we illustrate the energy-accuracy tradeoff of different NeuralFuse implementations. X-axis represents the percentage reduction in dynamic memory access energy at low-voltage settings (base model protected by NeuralFuse), compared to the bit-error-free (nominal) voltage. Y-axis represents the perturbed accuracy (evaluated at low voltage) with a $1\%$ bit error rate.

the original input and provide accurate inference. The pipeline of NeuralFuse is illustrated in Figure 1. NeuralFuse accepts the scenarios under access-limited neural networks (e.g., non-configurable hardware or remote access to cloud-based APIs) to protect the deployed models from making wrong predictions under low power. Specifically, we consider two practical access-limited scenarios: (a) *Relaxed Access*, where the model details are unknown but backpropagation through the black-box models is possible, and (b) *Restricted Access*, where models are unknown, and backpropagation is disallowed. For relaxed access, we train NeuralFuse via backpropagation, and for restricted access, we train NeuralFuse on a white-box surrogate model and transfer it to the restricted access models. To the best of our knowledge, this is the first study that leverages a learning-based method to address random bit errors for recovering accuracy in low-voltage and access-limited settings.

We summarize our **main contributions** as follows:

- We propose *NeuralFuse*, a novel learning-based input transformation framework to enhance the accuracy of DNNs subject to random bit errors caused by undervolting. NeuralFuse is model-agnostic because it operates in a plug-and-play manner at the data input and it does not require re-training the deployed DNN model.

- We consider two practical access-limited scenarios for neural network inference: *Relaxed Access* and *Restricted Access*. In the former setting, we use gradient-based methods to train the Neural-Fuse module. In the latter setting, we use a white-box surrogate model to train NeuralFuse and show its high transferability to other types of DNN architectures.

- We conduct extensive experiments on various combinations of DNN models (ResNet18, ResNet50, VGG11, VGG16, and VGG19), datasets (CIFAR-10, CIFAR-100, GTSRB, and ImageNet-10), and NeuralFuse implementations with different architectures and sizes. The results show that NeuralFuse can consistently increase the perturbed accuracy (accuracy evaluated under random bit errors in weights) by up to $57\%$, while simultaneously saving the overall SRAM memory access energy by up to $24\%$, based on the realistic characterization of bit cell failures for a given memory array in a low-voltage regime inducing a $0.5\%/1\%$ of bit error rate.

- We demonstrate NeuralFuse's advantages in various scenarios. The experimental results show that NeuralFuse has high transferability (adaptability to unseen base models), versatility (capable of recovering low-precision quantization loss, as well as bit errors), and competitiveness (achieving state-of-the-art performance).

## 2   RELATED WORK AND BACKGROUND

**Software-based Energy Saving Strategies.** Recent studies have proposed reducing energy consumption from a software perspective. For instance, quantization techniques have been proposed to

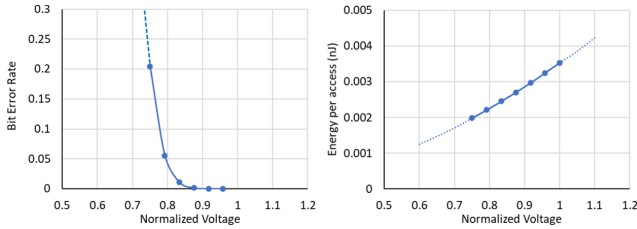

Figure 2: **Left**: Bit error rate. **Right**: Dynamic energy per memory access versus voltage for SRAM arrays in Chandramoorthy et al. (2019). The x-axis shows voltages normalized with respect to the minimum bit error-free voltage ($V_{min}$).

reduce the precision of storing model weights and decrease total memory storage (Gong et al., 2014; Rastegari et al., 2016; Wu et al., 2016). On the other hand, Yang et al. (2017) proposed energy-aware pruning on each layer and finetunes the weights to maximize the final accuracy. Yang et al. have also proposed several ways to reduce the energy consumption of DNN. For example, they proposed ECC, a DNN compression framework, that compresses a DNN model to meet the given energy constraint (Yang et al., 2019a); they have also proposed to compress DNN models via joint pruning and quantization (Yang et al., 2020). Besides, it is also feasible to treat energy constraint as an optimization problem during DNN training to reduce energy consumption and maximize training accuracy (Yang et al., 2019b). However, these methods focus on changing either the model architectures or model weights to reduce energy consumption, which is orthogonal to our approach with NeuralFuse, which serves as an add-on module for any given model.

**Hardware-based Energy Saving Strategies.** Existing works have studied improving energy efficiency by designing specific hardware. Several works have studied the undervolting of DNN accelerators and proposed methods to maintain accuracy in the presence of bit errors. For instance, Reagen et al. (2016) proposed an SRAM fault mitigation technique that rounds the faulty weights into zeros to avoid the degradation of the prediction accuracy. Srinivasan et al. (2016) proposed to store the sensitive MSBs (most significant bits) in robust SRAM cells to preserve accuracy. Chandramoorthy et al. (2019) proposed dynamic supply voltage boosting to a higher voltage to improve the resilience of the memory access operation. On the other hand, Stutz et al. (2021) considered a learning-based approach that tries to find models that are robust to bit errors. The paper discusses several techniques to improve the robustness, such as quantization methods, weight clipping, random bit error training, and adversarial bit error training. The authors observe that the combination of quantization, weight clipping, and adversarial bit error training achieves excellent performance in their experiments. However, they also admitted that the training process for that is sensitive to hyperparameter settings, and hence it might come with a challenging training procedure.

We argue that the methods mentioned above are not easy to implement or not suitable for real-world scenarios in *access-limited* settings. For example, the weights of DNN models packed on embedded systems may not be configurable or updatable. Therefore, model retraining (e.g., Stutz et al. (2021)) is not a viable option. Moreover, the training of DNNs is already a tedious and time-consuming task. Adding error-aware training during training may further increase the training complexity and introduce challenges in hyperparameter search as identified in previous literature. Özdenizci & Legenstein (2022) also note that this error-aware training was also found ineffective for large DNNs with millions of bits. Our proposed *NerualFuse* spares the need for model retraining by attaching a trainable input transformation function parameterized by a relatively small DNN as an add-on module to any DNN model as is.

**SRAM Bit Errors in DNNs.** Low-voltage-induced memory bit cell failures can cause bit flips from 0 to 1 and vice versa. In practice, SRAM memory bit errors increase exponentially when the supply voltage is scaled below $V_{min}$, which is the minimum voltage required to avoid bit errors. This phenomenon has been well studied in previous literature, such as the work by Chandramoorthy et al. (2019) and Ganapathy et al. (2017). As shown in Figure 2, for an SRAM array of size $512 \times 64$ bits and 14nm technology node, the number of bit errors increases as the voltage scales down. The corresponding dynamic energy per read access of SRAM is shown on the right, measured at each voltage at a constant frequency. In this example, accessing the SRAM at $0.83V_{min}$ leads to a 1% bit error rate, and at the same time, the dynamic energy per access is reduced by approximately 30%.

This can lead to inaccurate inferences in DNNs, particularly when bit flips occur at the MSBs. However, improving the robustness to bit errors allows us to lower the voltage and exploit the resulting energy savings.

It has been observed that bit cell failures for a given memory array are randomly distributed and independent of each other. That is, the spatial distribution of bit flips could be assumed random, as it is generally different between arrays, even within a chip and between different chips. In this paper, we follow the methodology in Chandramoorthy et al. (2019) and model bit errors in a memory array of a given size by generating a random distribution of bit errors with equal likelihood of 0-to-1 and 1-to-0 bit flips. Specifically, we assume that the model weights are quantized to 8-bit precision (i.e., from 32-bit floats to 8-bit integers), and randomly distributed bit errors are injected into the quantized 2's complement representation of weights to generate perturbed models. Please refer to Section 4.1 for more implementation details.

## 3 NEURALFUSE: FRAMEWORK AND ALGORITHMS

### 3.1 ERROR-RESISTANT INPUT TRANSFORMATION

As illustrated in Figure 1, to overcome the drawback of performance degradation in low-voltage regimes, we propose a novel trainable input transformation module parametrized by a relatively small DNN, *NeuralFuse*, to mitigate the accuracy-energy trade-off for model inference. The rationale is to use a specially designed loss function and training scheme to derive the NeuralFuse and apply it to the input data such that the transformed inputs will become robust to low-voltage-induced bit errors.

Consider the input $\mathbf{x}$ sampled from the data distribution $\mathcal{X}$ and a model $M_p$ with random bit errors on its weights (which is called a *perturbed* model). When there are no bit errors (i.e., the normal-voltage settings), the perturbed model reduces to a nominal deterministic model denoted by $M_0$. NeuralFuse aims to ensure the perturbed model $M_p$ can make correct inferences on the transformed inputs as well as retain consistent results of $M_0$ in regular (normal-voltage) settings. In order to adapt to different data characteristics, NeuralFuse ($\mathcal{F}$) is designed to be input-aware, which can be formally defined as:

$$\mathcal{F}(\mathbf{x}) = \text{clip}_{[-1,1]}\big(\mathbf{x} + \mathcal{G}(\mathbf{x})\big), \tag{1}$$

where $\mathcal{G}(\mathbf{x})$ is a "generator" (i.e., an input transformation function) that can generate a perturbation based on the input $\mathbf{x}$. The NeuralFuse transformed data $\mathcal{F}(\mathbf{x})$ will be passed to the deployed model ($M_0$ or $M_p$) for final inference. Without loss of generality, we assume the transformed input lies within a scaled input range $\mathcal{F}(\cdot) \in [-1,1]^d$, where $d$ is the (flattened) dimension of $\mathbf{x}$.

### 3.2 TRAINING OBJECTIVE AND OPTIMIZER

To train the generator $\mathcal{G}(\cdot)$, which should ensure the correctness of both the perturbed model $M_p$ and the clean model $M_0$, we parameterize $\mathcal{G}(\cdot)$ by a neural network and design the following training objective function:

$$\arg\max_{\mathcal{W}_{\mathcal{G}}} \log \mathcal{P}_{M_0}(y|\mathcal{F}(\mathbf{x}; \mathcal{W}_{\mathcal{G}})) + \lambda \cdot \mathbf{E}_{M_p \sim \mathcal{M}_p}[\log \mathcal{P}_{M_p}(y|\mathcal{F}(\mathbf{x}; \mathcal{W}_{\mathcal{G}}))], \tag{2}$$

where $\mathcal{W}_{\mathcal{G}}$ is the set of trainable parameters for $\mathcal{G}$, $y$ is the ground-truth label of $\mathbf{x}$, $\mathcal{P}_M$ denotes the likelihood of $y$ computed by a model $M$ given a transformed input $\mathcal{F}(\mathbf{x}; \mathcal{W}_{\mathcal{G}})$, $\mathcal{M}_p$ is the distribution of the perturbed models inherited from the clean model $M_0$ under a $p\%$ random bit error rate, and $\lambda$ is a hyperparameter that balances the importance between the nominal and perturbed models.

The training objective function can be readily converted to a loss function (*loss*) that evaluates the cross-entropy between the ground-truth label $y$ and the prediction $\mathcal{P}_M(y|\mathcal{F}(\mathbf{x}; \mathcal{W}_{\mathcal{G}}))$. That is, the total loss function becomes

$$Loss_{Total} = loss_{M_0} + \lambda \cdot loss_{\mathcal{M}_p}. \tag{3}$$

To optimize the loss function entailing the evaluation of the loss term $loss_{\mathcal{M}_p}$ on randomly perturbed models, our training process is inspired by the EOT (Expectation Over Transformation) attacks (Athalye et al., 2018). EOT attacks aim to find a robust adversarial example against a variety of

image transformations. Based on the idea, we propose a new optimizer for solving equation 3, which we call Expectation Over Perturbed Models (EOPM). EOPM-trained generators can generate error-resistant input transformations and mitigate the inherent bit errors. However, it is computationally impossible to enumerate all possible perturbed models with random bit errors, and the number of realizations for perturbed models is limited by the memory constraint of GPUs used for training. In practice, we only take $N$ perturbed models for each iteration to calculate the empirical average loss, i.e.,

$$Loss_N = \frac{loss_{M_{p_1}} + ... + loss_{M_{p_N}}}{N}, \tag{4}$$

where $N$ is the number of simulated perturbed models $\{M_{p_1}, \ldots, M_{p_N}\}$ under random bit errors to calculate the loss. Therefore, the gradient used to update the generator can be calculated as follows:

$$\frac{\partial Loss_{Total}}{\partial \mathcal{W}_\mathcal{G}} = \frac{\partial loss_{M_0}}{\partial \mathcal{W}_\mathcal{G}} + \frac{\lambda}{N} \cdot \left( \frac{\partial loss_{M_{p_1}}}{\partial \mathcal{W}_\mathcal{G}} + ... + \frac{\partial loss_{M_{p_N}}}{\partial \mathcal{W}_\mathcal{G}} \right). \tag{5}$$

In our implementation, we find that $N = 10$ can already deliver stable performance, and there is little gain in using a larger value. The ablation study for different $N$ can be found in Appendix F.

### 3.3 Training Algorithm

Algorithm 1 in Appendix A summarizes the training steps of NeuralFuse. We split the training data $\mathcal{X}$ into $B$ mini-batches for training the generator in each epoch. For each mini-batch, we first feed these data into $\mathcal{F}(\cdot)$ to get the transformed inputs. Also, we simulate $N$ perturbed models using a $p\%$ random bit error rate, denoted by $M_{p_1}, ..., M_{p_N}$, from $\mathcal{M}_p$. Then, the transformed inputs are fed into these $N$ perturbed models and the clean model $M_0$ to calculate their losses and gradients. Finally, NeuralFuse parameters $\mathcal{W}_\mathcal{G}$ are updated based on the gradient obtained by EOPM.

## 4 Experiments

In this section, we present the experimental setup and results of NeuralFuse on different datasets and various architectures. In addition, we also provide the visualization results, detailed analysis, and ablation studies to better understand the properties of NeuralFuse.

### 4.1 Experiment Setups

**Datasets.** We evaluate NeuralFuse on four different datasets: CIFAR-10 (Krizhevsky & Hinton, 2009), CIFAR-100 (Krizhevsky & Hinton, 2009), GTSRB (Stallkamp et al., 2012), and ImageNet-10 (Deng et al., 2009). CIFAR-10 consists of ten classes, with 50,000 training images and 10,000 for testing. Similarly, CIFAR-100 consists of 100 classes, with 500 training images and 100 testing images in each class. GTSRB (German Traffic Sign Recognition Benchmark) is a dataset that contains 43 classes with 39,209 training images and 12,630 testing images. Similar to CIFAR-10 and CIFAR-100, we resize GTSRB into 32×32×3 in our experiment. For ImageNet-10, we chose the same ten categories as Huang et al. (2022), and there are 13,000 training images and 500 test images, which are cropped into 224×224×3. Due to the space limit, the CIFAR-100 results are given in Appendix.

**Base Models.** We select several common architectures for our base models: ResNet18, ResNet50 (He et al., 2016), VGG11, VGG16, and VGG19 (Simonyan & Zisserman, 2015). In order to meet the setting of deploying models on a chip, all of our based models are trained by quantization-aware training following Stutz et al. (2021).

**NeuralFuse Generators.** The architecture of the NeuralFuse generator ($\mathcal{G}$) is based on the Encoder-Decoder structure. We design and compare three types of generators, namely Convolution-based, Deconvolution-based, and UNet-based. For each type, we also consider large(L)/small(S) network sizes. Both Convolution-based and Deconvolution-based variants will follow a similar architecture for ease of comparison. Also, the generators were trained with quantization-aware training. More details are given in Appendix B.

- **Convolution-based (Conv)**: We use Convolution with MaxPool layers for the encoder, and Convolution with UpSample layers for the decoder. The architecture is inspired by Nguyen & Tran (2020).

- **Deconvolution-based (DeConv)**: We use Convolution with MaxPool layers for the encoder, and Deconvolution layers for the decoder.
- **UNet-based (UNet)**: UNet (Ronneberger et al., 2015) is known to attain strong performance on image segmentation. We use Convolution with MaxPool layers for the encoder, and Deconvolution layers for the decoder.

**Energy Consumption Calculation.** Chen et al. (2016) has shown that energy consumption in the SRAM part (both buffer and array) consumes a large amount of total system energy consumption. In this paper, we focus on the resilience to low-voltage-induced bit errors in model weights, and our reported energy consumption in Figure 1 is based on the product of the total number of SRAM memory accesses in a systolic-array-based CNN accelerator and the dynamic energy per read access at a given voltage. We assume that there is no bit error on NeuralFuse, as we consider it as an add-on, data preprocessing module that could also be performed by the general-purpose core. Therefore, when implemented in the accelerator, NeuralFuse computation is performed at a higher error-free voltage with a dynamically scaled voltage supply to the memories. We report the reduction in overall weight memory energy consumption (i.e., NeuralFuse + Base Model under $p\%$ bit error rate) with respect to the unprotected base model in the regular voltage mode (i.e., $0\%$ bit error rate and without NeuralFuse). Note that we do not report the overall/end-to-end energy consumption of the accelerator during computation, as end-to-end power savings will depend on various factors, such as the memory power as a fraction of the total hardware power, whether computation logic is also running at low voltage/frequency, or the use of multiple voltage domains in the accelerator.

To obtain the number of memory accesses, we used the SCALE-SIM simulator (Samajdar et al., 2020), and our chosen configuration simulates an output-stationary dataflow and a $32\times32$ systolic array with 256KB of weight memory. We obtained the dynamic energy per read access of the SRAM at the minimum voltage ($V_{min}$) and at the voltage corresponding to a $1\%$ bit error rate ($V_{ber} \approx 0.83V_{min}$) from Cadence ADE Spectre simulations, both at the same clock frequency. Please refer to Appendix C for more details.

**Relaxed and Restricted Access Settings.** We consider two scenarios (relaxed/restricted access) in our experiments. For relaxed access, the information of the base model is not entirely transparent but it allows obtaining gradients from the black-box model through backpropagation. Therefore, this setting allows direct training of NeuralFuse with the base model using EOPM. On the other hand, for restricted access, only the inference function is allowed for the base model. Therefore, we train NeuralFuse by using a white-box surrogate base model and then transfer the generator to the access-restricted model.

**Computing Resources.** Our experiments are performed using 8 Nvidia Tesla V100 GPUs, and are implemented with PyTorch. NeuralFuse generally takes 150 epochs to converge, and its training time is similar for the same base model. On both CIFAR-10 and CIFAR-100 datasets, the average training times were 17 hours (ResNet18), 50 hours (ResNet50), 9 hours (VGG11), 13 hours (VGG16), and 15 hours (VGG19). For GTSRB, the average training times were 9 hours (ResNet18), 27 hours (ResNet50), 5 hours (VGG11), 7 hours (VGG16), and 8 hours (VGG19). For ImageNet-10, the average training times were 32 hours (ResNet18), 54 hours (ResNet50), 50 hours (VGG11), 90 hours (VGG16), and 102 hours (VGG19).

## 4.2 PERFORMANCE EVALUATION ON RELAXED ACCESS SETTING

The experimental results of the Relaxed Access setting are shown in Figure 3. We train and test NeuralFuse with various base models (ResNet18, ResNet50, VGG11, VGG16, and VGG19). Two power settings have been considered: *nominal voltage* (no bit error) and *low voltage* (random bit errors), and the corresponding bit error rate (B.E.R.) due to low voltage is $1\%$ (CIFAR-10, GTSRB) and $0.5\%$ (ImageNet-10). The B.E.R. of ImageNet-10 is lower because the pre-trained models have more parameters than CIFAR-10 and GTSRB. For each experiment, we sample $N = 10$ perturbed models (independent from training) for evaluation and report the mean and standard deviation of the test accuracy. In the following, clean accuracy (**C.A.**) means that the model is measured at nominal voltage, and perturbed accuracy (**P.A.**) means that it is measured at low voltage.

For CIFAR-10 and GTSRB, we observe that large generators like ConvL and UNetL can significantly recover the perturbed accuracy in the range of $41\%$ to $63\%$ on ResNet18, VGG11, VGG16,

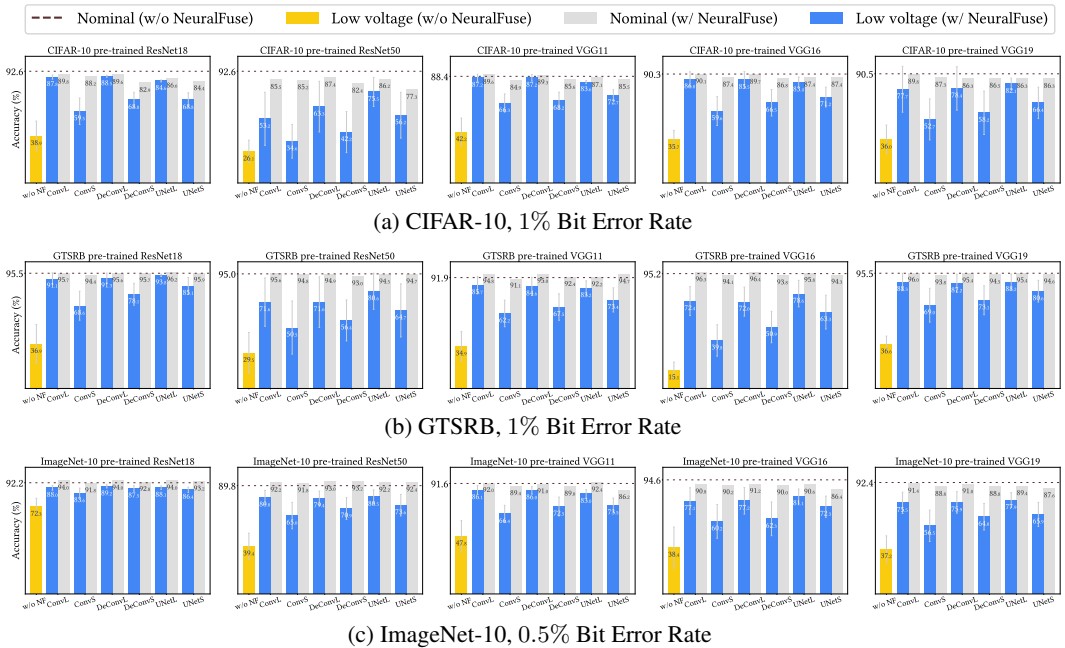

Figure 3: Relaxed Access setting: Test accuracy (%) of different pre-trained models with or without NeuralFuse, compared at nominal voltage (0% bit error rate) or low voltage (with specified bit error rates). The results demonstrate that NeuralFuse consistently recovers perturbation accuracy.

and VGG19. For ResNet50, the recover percentage is slightly worse than other base models, but it can attain up to 51% recover percentage on GTSRB. On the other hand, the recover percentage based on small generators like DeConvS are worse than larger generators. This can be explained by the better ability to learn error-resistant generators for larger-sized networks (though they may consume more energy). For ImageNet-10, using larger generators can also gain better performance recovery on perturbed accuracy. This also demonstrates that NeuralFuse can still work well even with large input sizes and is applicable to different datasets.

### 4.3 TRANSFERABILITY FOR RESTRICTED ACCESS SETTING

In the Restricted Access scenario, we train NeuralFuse generators on a white-box surrogate base model and transfer it to other black-box base models. The experimental results are shown in Table 1. We adopt ResNet18 and VGG19 as the white-box surrogate (source) models for training the generators under a 1.5% bit error rate. For the generators, we choose ConvL and UNetL as they obtain better performance in Figure 3.

From Table 1 we can find that transferring from a larger B.E.R. like 1.5% can give strong resilience to a smaller B.E.R. like 1% or 0.5%. We also find that using VGG19 as a surrogate model with UNet-based generators like UNetL can give better recovery than other combinations. On the other hand, in some cases, we observe that if we transfer between the same source and target models (but with different B.E.R. for training and testing), the performance may outperform the original relaxed-access results. For instance, when transferring VGG19 with UNetL under a 1.5% B.E.R. to VGG19 or VGG11 under a 0.5% B.E.R., the accuracy is 85.86% v.s. 84.99% for VGG19 (original), and 84.81% v.s. 82.42% for VGG11 (original), respectively. We conjecture that the generators trained on a larger B.E.R. can actually cover the error patterns of a smaller B.E.R., and even help improve generalization under a smaller B.E.R. These findings show great promise for recovering the accuracy of access-limited base models in low-voltage settings.

### 4.4 ENERGY-ACCURACY TRADEOFF

We report the total dynamic energy consumption as the total number of SRAM accesses times the dynamic energy of a single SRAM access. Specifically, we used SCALE-SIM to calculate the total weight memory access (T.W.M.A.), which can be found in Table 6 in Appendix D. In Table 2, we

Table 1: Restricted Access setting: Transfer results trained with $1.5\%$ bit error rate on CIFAR-10.

| S.M. | T.M. | B.E.R. | C.A. | P.A. | ConvL (1.5%) | | | UNetL (1.5%) | | |
|---|---|---|---|---|---|---|---|---|---|---|
| | | | | | C.A. (NF) | P.A. (NF) | R.P. | C.A. (NF) | P.A. (NF) | R.P. |
| ResNet18 | ResNet18 | 1% | 92.6 | $38.9 \pm 12.4$ | 89.8 | $89.0 \pm 0.5$ | 50.1 | 85.8 | $85.2 \pm 0.5$ | 46.3 |
| | | 0.5% | | $70.1 \pm 11.6$ | | $89.6 \pm 0.2$ | 19.5 | | $85.7 \pm 0.2$ | 15.6 |
| | ResNet50 | 1% | 92.6 | $26.1 \pm\ \ 9.4$ | 89.2 | $36.1 \pm\ 18$ | 10.0 | 84.4 | $38.9 \pm\ 16$ | 12.8 |
| | | 0.5% | | $61.0 \pm 10.3$ | | $74.1 \pm\ 10$ | 13.1 | | $72.7 \pm 4.6$ | 11.7 |
| | VGG11 | 1% | 88.4 | $42.2 \pm 11.6$ | 86.3 | $59.2 \pm\ 10$ | 17.0 | 82.3 | $69.8 \pm 7.5$ | 27.6 |
| | | 0.5% | | $63.6 \pm\ \ 9.3$ | | $78.9 \pm 4.9$ | 15.3 | | $77.0 \pm 4.0$ | 13.4 |
| | VGG16 | 1% | 90.3 | $35.7 \pm\ \ 7.9$ | 89.4 | $62.2 \pm\ 18$ | 26.5 | 84.7 | $68.9 \pm\ 14$ | 33.2 |
| | | 0.5% | | $66.6 \pm\ \ 8.1$ | | $83.4 \pm 5.5$ | 16.8 | | $80.5 \pm 5.9$ | 13.9 |
| | VGG19 | 1% | 90.5 | $36.0 \pm 12.0$ | 89.8 | $49.9 \pm\ 23$ | 13.9 | 85.0 | $55.1 \pm\ 17$ | 19.1 |
| | | 0.5% | | $64.2 \pm 12.4$ | | $81.8 \pm 8.5$ | 17.6 | | $78.5 \pm 6.8$ | 14.3 |
| VGG19 | ResNet18 | 1% | 92.6 | $38.9 \pm 12.4$ | 88.9 | $62.6 \pm\ 13$ | 23.7 | 85.0 | $72.3 \pm\ 11$ | 33.4 |
| | | 0.5% | | $70.1 \pm 11.6$ | | $84.2 \pm 7.2$ | 14.1 | | $82.1 \pm 2.2$ | 12.0 |
| | ResNet50 | 1% | 92.6 | $26.1 \pm\ \ 9.4$ | 88.8 | $37.9 \pm\ 18$ | 11.8 | 85.2 | $46.7 \pm\ 17$ | 20.6 |
| | | 0.5% | | $61.0 \pm 10.3$ | | $76.6 \pm 7.8$ | 15.6 | | $78.3 \pm 3.7$ | 17.3 |
| | VGG11 | 1% | 88.4 | $42.2 \pm 11.6$ | 88.9 | $76.0 \pm 6.1$ | 33.8 | 85.5 | $81.9 \pm 3.9$ | 39.7 |
| | | 0.5% | | $63.6 \pm\ \ 9.3$ | | $85.9 \pm 2.6$ | 22.3 | | $84.8 \pm 0.5$ | 21.2 |
| | VGG16 | 1% | 90.3 | $35.7 \pm\ \ 7.9$ | 89.0 | $76.5 \pm 9.0$ | 40.8 | 85.9 | $79.2 \pm 7.5$ | 43.5 |
| | | 0.5% | | $66.6 \pm\ \ 8.1$ | | $87.7 \pm 0.7$ | 21.1 | | $84.7 \pm 0.9$ | 18.1 |
| | VGG19 | 1% | 90.5 | $36.0 \pm 12.0$ | 89.1 | $80.2 \pm\ 12$ | 44.2 | 86.3 | $84.3 \pm 1.2$ | 48.3 |
| | | 0.5% | | $64.2 \pm 12.4$ | | $88.8 \pm 0.4$ | 24.6 | | $85.9 \pm 0.3$ | 21.7 |

[Note] S.M.: *source model, used for training generators*, T.M.: *target model, used for testing generators*, B.E.R.: *the bit error rate of the target model*, C.A. (%): *clean accuracy*, P.A. (%): *perturbed accuracy*, NF: *NeuralFuse*, and R.P.: *total recover percentage of P.A. (NF) v.s. P.A.*

report the percentage of energy savings at voltages that would yield a $1\%$ bit error rate for various base model and generator combinations. The formula of the energy saving percentage (ES, %) is defined as:

$$\text{ES} = \frac{\text{Energy}_{\text{nominal voltage}} - \left(\text{Energy}_{\text{low-voltage-regime}} + \text{Energy}_{\text{NeuralFuse at nominal voltage}}\right)}{\text{Energy}_{\text{nominal voltage}}} \times 100\%.$$

Table 2: The energy saving percentage (%) for different combinations of base models and Neural-Fuse.

| Base Model | ConvL | ConvS | DeConvL | DeConvS | UNetL | UNetS |
|---|---|---|---|---|---|---|
| ResNet18 | 19.0 | 29.1 | 21.2 | 27.5 | 24.0 | 28.9 |
| ResNet50 | 25.4 | 29.9 | 26.4 | 29.2 | 27.7 | 29.9 |
| VGG11 | 6.6 | 27.5 | 11.2 | 24.1 | 17.1 | 27.2 |
| VGG16 | 17.1 | 28.9 | 19.7 | 27.0 | 23.0 | 28.7 |
| VGG19 | 20.3 | 29.7 | 22.3 | 27.8 | 24.8 | 29.1 |

Also, as shown in Figure 1(b), when utilizing ResNet18 as a base model, NeuralFuse can recover model accuracy by $20\% \sim 49\%$ while saving $19\% \sim 29\%$ energy. More results are in Appendix D, and we have provided Multiply-Accumulate operations-based energy-saving results in Appendix E.

## 4.5 MODEL SIZE AND EFFICIENCY OF NEURALFUSE

To provide a full performance characterization of NeuralFuse, we analyze the relationship between the final recovery of each base model and generators of varying parameter counts. The *efficiency ratio* is defined as the recover percentage in perturbed accuracy divided by the parameter count of NeuralFuse. We compare the efficiency ratio for all NeuralFuse generators (trained on CIFAR-10) in Table 3. The results show that UNet-based generators have better efficiency than both Convolution-based and Deconvolution-based ones per million parameters.

## 4.6 NEURALFUSE ON REDUCED-PRECISION QUANTIZATION

Here we explore the robustness of NeuralFuse to low-precision quantization on model weights. Uniform quantization is the de-facto method for quantizing model weights (Gholami et al., 2022). However, it is possible to cause an accuracy drop due to precision loss. As a bit-error-oriented protector, we seek to understand whether NeuralFuse could also make a recovery to mitigate the model's accuracy drop in this scope. We conducted an experiment that uniformly quantized the model weights to

Table 3: The efficiency ratio (%) for all NeuralFuse generators.

| Base Model | B.E.R. | NeuralFuse | | | | | |
| | | ConvL | ConvS | DeConvL | DeConvS | UNetL | UNetS |
|---|---|---|---|---|---|---|---|
| ResNet18 | 1% | 67.5 | 182 | 76.6 | 190.7 | 94.5 | 245.9 |
| | 0.5% | 24.7 | 73.3 | 30.7 | 62.5 | 33.6 | 88.3 |
| ResNet50 | 1% | 37.4 | 75.1 | 57.4 | 102.7 | 102.3 | 248.4 |
| | 0.5% | 35.2 | 108.7 | 40.4 | 92.5 | 47.4 | 124.6 |
| VGG11 | 1% | 62.3 | 212.9 | 69.5 | 165.8 | 92.0 | 251.7 |
| | 0.5% | 32.3 | 96.3 | 35.8 | 77.2 | 38.9 | 100.7 |
| VGG16 | 1% | 69.6 | 211.2 | 76.9 | 196.5 | 98.8 | 292.9 |
| | 0.5% | 30.3 | 98.1 | 33.0 | 75.3 | 40.6 | 113 |
| VGG19 | 1% | 57.6 | 147.5 | 65.5 | 141.6 | 95.4 | 250.8 |
| | 0.5% | 33.0 | 91.0 | 37.5 | 70.2 | 43.1 | 106.4 |

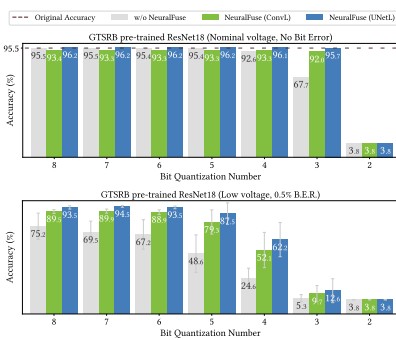

Figure 4: Reduced-precision accuracy

a lower bit precision and reported the resulting accuracy. Specifically, we apply symmetric uniform quantization on our base models with different numbers of bits to induce precision loss. Therefore, the quantized weight $\mathbf{W_q}$ (integer) is defined as $\mathbf{W_q} = \lfloor \frac{\mathbf{W}}{\mathbf{s}} \rceil$, where $\mathbf{W}$ denotes the original model weight (full precision), $s = \frac{\max|\mathbf{W}|}{2^{b-1}-1}$ denotes the quantization scale parameter, and $b$ is the precision (number of bits) that used to quantize the models. In addition, we considered the possibility of random bit errors (the low-voltage setup) in low-precision regimes.

We use GTSRB pre-trained ResNet18 as an example to evaluate two NeuralFuse generators (ConvL and UNetL, trained with $0.5\%$ B.E.R.), and vary the precision $b$ from 8 bits to 2 bits (integer). The result is shown in Figure 4. We find that when $b > 3$ bits, NeuralFuse can take effect to recover the accuracy in both scenarios. When $b = 3$, while NeuralFuse can still handle the bit-error-free model (top panel), it exhibits a limited ability to recover the random bit-error case (bottom panel). We find the result encouraging because the observed robustness to reduced-precision inference is an emergent ability of NeuralFuse. That is, NeuralFuse was only trained with random bit errors, but it demonstrated high accuracy to unseen bit quantization errors. This experiment showcases NeuralFuse's potential in protecting against accuracy drops caused by different sources of bit errors. More experimental results of different base models and datasets can be found in Appendix I.

## 4.7 EXTENDED ANALYSIS

We highlight some key findings from the additional results in Appendix. In Appendix F, we compare NeuralFuse to the simple baseline of learning a universal input perturbation. We find that the baseline is much worse than NeuralFuse, which validates the necessity of adopting input-aware transformation for learning error-resistant data representations in low-voltage scenarios. In Appendix H, we find that ensemble training of white-box surrogate base models can further improve the transferability of NeuralFuse in the restricted-access setting. In Appendix J and Appendix K, we present visualization results of data embeddings and transformed inputs via NeuralFuse. In Appendix M, we show that NeuralFuse can further recover the accuracy of a base model trained with adversarial weight perturbation in the low-voltage setting.

## 5 CONCLUSION

In this paper, we propose NeuralFuse, the first non-intrusive post-hoc protection module for model inference against bit errors induced by low voltage. NeuralFuse is particularly suited for practical machine deployment settings where access to the base model is limited or relaxed. The design of NeuralFuse includes a novel loss function and a new optimizer named EOPM to handle simulated randomness in perturbed models. Our comprehensive experimental results and analysis show that NeuralFuse can significantly recover test accuracy (by up to $57\%$) while simultaneously enjoying up to $24\%$ reduction in memory access energy. Furthermore, NeuralFuse demonstrates high transferability (to access-constrained models), and versatility (e.g., robustness to low-precision quantization). Our results show that NeuralFuse provides significant improvements in mitigating the energy-accuracy tradeoff of neural network inference in low-voltage regimes and sheds new insights on green AI technology. Our future work includes extending our study to other neural network architectures and modalities, such as transformer-based language models.

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

APPENDIX

In the appendix, we provide more implementation details for our method, experimental results on more datasets and settings, ablation studies, and qualitative analysis. The appendices cover the following:

- **Implementation Details:** NeuralFuse Training Algorithm (Sec. A), NeuralFuse Generator (Sec. B), SCALE-SIM (Sec. C)
- **Qualitative Studies:** Energy-Accuracy Tradeoff (Sec. D), Model Parameters and MAC Values (Sec. E), Data Embeddings Visualization (Sec. J), Transformed Inputs Visualization (Sec. K), Latency Reports (Sec. N)
- **Additional Experimental Results:** Ablation Studies (Sec. F), Relaxed Access (Sec. G), Restricted Access (Sec. H), Reduced-precision Quantization (Sec. I), Adversarial Training (Sec. L), Adversarial Weight Perturbation (Sec. M)

Our code can be found at https://anonymous.4open.science/r/neuralfuse/.

## A  TRAINING ALGORITHM OF NEURALFUSE

---

**Algorithm 1** Training steps for NeuralFuse

---

**Input**: Base model $M_0$; Generator $\mathcal{G}$; Training data samples $\mathcal{X}$; Distribution of the perturbed models $\mathcal{M}_p$; Number of perturbed models $N$; Total training iterations $T$
**Output**: Optimized parameters $\mathcal{W}_\mathcal{G}$ for the Generator $\mathcal{G}$

1: **for** $t = 0, ..., T - 1$ **do**
2:     **for all** mini-batches $\{\mathbf{x}, y\}_{b=1}^B \sim \mathcal{X}$ **do**
3:         Create transformed inputs $\mathbf{x}_t = \mathcal{F}(\mathbf{x}) = \text{clip}_{[-1,1]}\left(\mathbf{x} + \mathcal{G}(\mathbf{x})\right)$.
4:         Sample $N$ perturbed models $\{M_{p_1}, ..., M_{p_N}\}$ from $\mathcal{M}_p$ under $p\%$ random bit error rate.
5:         **for all** $M_{p_i} \sim \{M_{p_1}, ..., M_{p_N}\}$ **do**
6:             Calculate the loss $loss_{p_i}$ based on the output of the perturbed model $M_{p_i}$. Then calculate the gradients $g_{p_i}$ for $\mathcal{W}_\mathcal{G}$ based on $loss_{p_i}$.
7:         **end for**
8:         Calculate the loss $loss_0$ based on the output of the clean model $M_0$. Then calculate the gradients $g_0$ for $\mathcal{W}_\mathcal{G}$ based on $loss_0$.
9:         Calculate the final gradient $g_{final}$ using (5) based on $g_0$ and $g_{p_1}, ..., g_{p_N}$.
10:        Update $\mathcal{W}_\mathcal{G}$ using $g_{final}$.
11:     **end for**
12: **end for**

---

## B  IMPLEMENTATION DETAILS OF NEURALFUSE GENERATOR

We consider two main goals in designing the NeuralFuse Generator: 1) efficiency (so the overall energy overhead is decreased) and 2) robustness (so that it can generate robust patterns on the input image and overcome the random bit flipping in subsequent models). Accordingly, we choose to utilize an encode-decoder architecture in implementing the generator. The design of ConvL is inspired by Nguyen & Tran (2020), in which the authors utilize a similar architecture to design an input-aware trigger generator, and have demonstrated its efficiency and effectiveness. Furthermore, we attempted to enhance it by replacing the *Upsampling* layer with a *Deconvolution* layer, leading to the creation of DeConvL. The UNetL-based NeuralFuse draws inspiration from Ronneberger et al. (2015), known for its robust performance in image segmentation, and thus, we incorporated it as one of our architectures. Lastly, ConvS, DeConvS, and UNetS are *scaled-down* versions of the model designed to reduce computational costs and total parameters. The architectures of Convolutional-based and Deconvolutional-based are shown in Table 4, and the architecture of UNet-based generators is in Table 5. For the abbreviation used in the table, ConvBlock means the Convolution block, Conv means a single Convolution layer, DeConvBlock means the Deconvolution block, DeConv means a single Deconvolution layer, and BN means a Batch Normalization layer. We use learning rate $= 0.001$, $\lambda = 5$, and Adam optimizer. For CIFAR-10, GTSRB, and CIFAR-100, we set batch size $b = 25$ for each base model. For ImageNet-10, we set $b = 64$ for ResNet18, ResNet50 and VGG11, and $b = 32$ for both VGG16 and VGG19.

Table 4: Model architecture for both Convolution-based and Deconvolution-based generators. Each ConvBlock consists of a Convolution (kernel $= 3 \times 3$, padding $= 1$, stride $= 1$), a Batch Normalization, and a ReLU layer. Each DeConvBlock consists of a Deconvolution (kernel $= 4 \times 4$, padding $= 1$, stride $= 2$), a Batch Normalization, and a ReLU layer.

| ConvL | | ConvS | | DeConvL | | DeConvS | |
|---|---|---|---|---|---|---|---|
| Layers | #CHs | Layers | #CHs | Layers | #CHs | Layers | #CHs |
| (ConvBlock)×2, MaxPool | 32 | ConvBlock, Maxpool | 32 | (ConvBlock)×2, MaxPool | 32 | ConvBlock, Maxpool | 32 |
| (ConvBlock)×2, MaxPool | 64 | ConvBlock, Maxpool | 64 | (ConvBlock)×2, MaxPool | 64 | ConvBlock, Maxpool | 64 |
| (ConvBlock)×2, MaxPool | 128 | ConvBlock, Maxpool | 64 | (ConvBlock)×2, MaxPool, | 128 | ConvBlock, Maxpool | 64 |
| ConvBlock, UpSample, ConvBlock | 128 | ConvBlock, UpSample | 64 | ConvBlock | 128 | DeConvBlock | 64 |
| ConvBlock, UpSample, ConvBlock | 64 | ConvBlock, UpSample | 32 | DeConvBlock, ConvBlock | 64 | DeConvBlock | 32 |
| ConvBlock, UpSample, ConvBlock | 32 | ConvBlock, UpSample | 3 | DeConvBlock, ConvBlock | 32 | DeConv, BN, Tanh | 3 |
| Conv, BN, Tanh | 32 | Conv, BN, Tanh | 3 | Conv, BN, Tanh | 3 | | |

[Note] #CHs: *number of channels.*

Table 5: Model architecture for UNet-based generators. Each ConvBlock consists of a Convolution (kernel $= 3 \times 3$, padding $= 1$, stride $= 1$), a Batch Normalization, and a ReLU layer. Other layers, such as the Deconvolutional layer (kernel $= 2 \times 2$, padding $= 1$, stride $= 2$), are used in UNet-based models. For the final Convolution layer, the kernel size is set to 1.

| UNetL | | UNetS | |
|---|---|---|---|
| Layers | #Channels | Layers | #Channels |
| L1: (ConvBlock)×2 | 16 | L1: (ConvBlock)×2 | 8 |
| L2: Maxpool, (ConvBlock)×2 | 32 | L2: Maxpool, (ConvBlock)×2 | 16 |
| L3: Maxpool, (ConvBlock)×2 | 64 | L3: Maxpool, (ConvBlock)×2 | 32 |
| L4: Maxpool, (ConvBlock)×2 | 128 | L4: Maxpool, (ConvBlock)×2 | 64 |
| L5: DeConv | 64 | L5: DeConv | 32 |
| L6: Concat[L3, L5] | 128 | L6: Concat[L3, L5] | 64 |
| L7: (ConvBlock)×2 | 64 | L7: (ConvBlock)×2 | 32 |
| L8: DeConv | 32 | L8: DeConv | 16 |
| L9: Concat[L2, L8] | 64 | L9: Concat[L2, L8] | 32 |
| L10: (ConvBlock)×2 | 32 | L10: (ConvBlock)×2 | 16 |
| L11: DeConv | 16 | L11: DeConv | 8 |
| L12: Concat[L1, L11] | 32 | L12: Concat[L1, L11] | 16 |
| L13: (ConvBlock)×2 | 16 | L13: (ConvBlock)×2 | 8 |
| L14: Conv | 3 | L14: Conv | 3 |

## C    IMPLEMENTATION DETAILS OF SCALE-SIM

SCALE-SIM (Samajdar et al., 2020) is a systolic array based CNN simulator that can calculate the number of memory accesses and the total time in execution cycles by giving the specific model architecture and accelerator architectural configuration as inputs. In this paper, we use SCALE-SIM to calculate the weights memory access of 5 based models (ResNet18, ResNet50, VGG11, VGG16, VGG19), and 6 generators (ConvL, ConvS, DeConvL, DeConvS, UNetL, UNetS). While SCALE-SIM supports both Convolutional and Linear layers, it does not yet support Deconvolution layers. Instead, we try to approximate the memory costs of Deconvolution layers by Convolution layers. We change the input and output from Deconvolution into the output and input of the Convolution layers. Besides, we also change the stride into 1 when we approximate it. We also add padding for the convolution layers while generating input files for SCALE-SIM. In this paper, we only consider the energy saving on weights accesses, so we only take the value "SRAM Filter Reads" from the output of SCALE-SIM as the *total weights memory accesses* (T.W.M.A.) for further energy calculation.

# D  THE ENERGY-ACCURACY TRADEOFF UNDER 1% BIT ERROR RATE

In Table 6, we report the total weight memory access (T.W.M.A.) calculated by SCALE-SIM. We then showed the energy-accuracy tradeoff between all of the combinations of NeuralFuse and base models under a 1% of bit error rate in Figure 5.

Table 6: The total weights memory access calculated by SCALE-SIM.

| Base Model | ResNet18 | ResNet50 | VGG11 | VGG16 | VGG19 | - |
|---|---|---|---|---|---|---|
| T.W.M.A. | 2,755,968 | 6,182,144 | 1,334,656 | 2,366,848 | 3,104,128 | - |
| NeuralFuse | ConvL | ConvS | DeConvL | DeConvS | UNetL | UNetS |
| T.W.M.A. | 320,256 | 41,508 | 259,264 | 86,208 | 180,894 | 45,711 |

[Note] T.W.M.A.: *total weight memory access*.

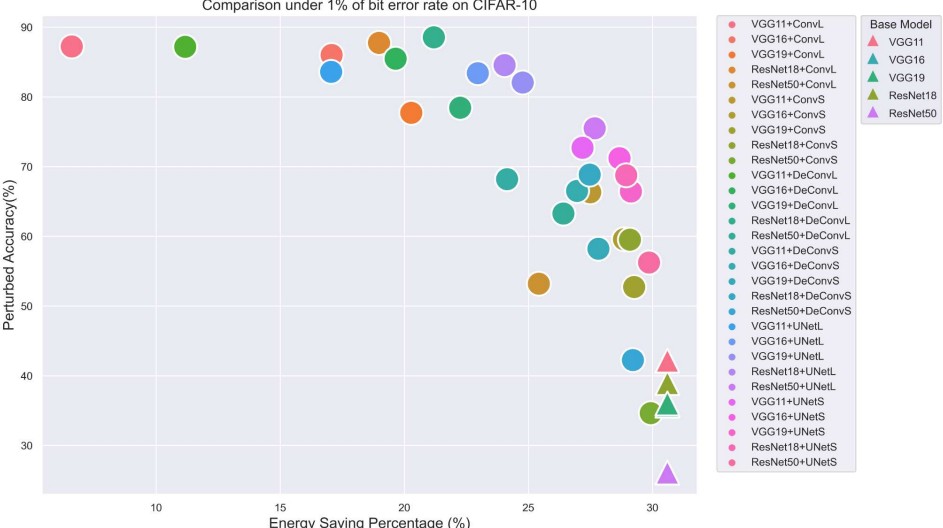

Figure 5: The energy-accuracy tradeoff of different NeuralFuse implementations with all CIFAR-10 pre-trained based models. X-axis represents the percentage reduction in dynamic memory access energy at low-voltage settings (base model protected by NeuralFuse), compared to the bit-error-free (nominal) voltage. Y-axis represents the perturbed accuracy (evaluated at low voltage) with a 1% bit error rate.

# E  MODEL PARAMETERS AND MAC VALUES

In addition to T.W.M.A., the model's parameters and MACs (multiply–accumulate operations) are common metrics in measuring the energy consumption of machine learning models. Yang et al. (2017) have also shown that the *energy consumption of computation* and *memory accesses* are both proportional to MACs, allowing us to estimate the overall (or end-to-end) energy consumption.

Here, we use the open-source package `ptflops` (Sovrasov, 2018-2023) to calculate the parameters and MAC values of all the base models and the NeuralFuse generators, in the same units as Bejnordi et al. (2020) used. The results are shown in Table 7. Note that we modified the base model architectures for ImageNet-10, as it has larger input sizes. For example, we use a larger kernel size = 7 instead of 3 in the first Convolution layer in ResNet-based models to enhance the learning abilities. Therefore, the parameters of base models are different between different datasets. For NeuralFuse generators, we utilize the same architectures for implementation (including ImageNet-10). As a result, our proposed NeuralFuse generators are generally smaller than base models, either on total model parameters or MAC values.

Table 7: Parameter counts and MACs for all base models and generators in this paper.

| | | Base Model | | | | | |
| --- | --- | --- | --- | --- | --- | --- | --- |
| | | ResNet18 | ResNet50 | VGG11 | VGG16 | VGG19 | - |
| Parameter | CIFAR-10 | 11,173,962 | 23,520,842 | 9,231,114 | 14,728,266 | 20,040,522 | - |
| | ImageNet-10 | 11,181,642 | 23,528,522 | 128,812,810 | 134,309,962 | 139,622,218 | |
| MACs | CIFAR-10 | 557.14M | 1.31G | 153.5M | 314.43M | 399.47M | - |
| | ImageNet-10 | 1.82G | 4.12G | 7.64G | 15.53G | 19.69G | |
| | | NeuralFuse | | | | | |
| | | ConvL | ConvS | DeConvL | DeConvS | UNetL | UNetS |
| Parameter | CIFAR-10 | 723,273 | 113,187 | 647,785 | 156,777 | 482,771 | 121,195 |
| | ImageNet-10 | | | | | | |
| MACs | CIFAR-10 | 80.5M | 10.34M | 64.69M | 22.44M | 41.41M | 10.58M |
| | ImageNet-10 | 3.94G | 506.78M | 3.17G | 1.1G | 2.03G | 518.47M |

**MACs-Based Energy Saving Calculation.** We can then use the MAC values to further approximate the end-to-end energy consumption of the whole model. Assume that all values are stored on SRAM and that a MAC represents single memory access. The corresponding MACs-based energy saving percentage (MAC-ES, %) can be derived from Eq. 6 (c.f. Sec. 4.4), and results can be found in Table 8. We can observe that most combinations can save a large amount of energy, except that VGG11 with two larger NeuralFuse (ConvL and DeConvL) may increase the total energy. These results are consistent with the results reported in Table 2. In addition, we also showed the MACs-based energy-accuracy tradeoff between all of the combinations of NeuralFuse and base models under a 1% of bit error rate in Figure 6.

$$\text{MAC-ES} = \frac{\text{MACs}_{\text{base model}} \cdot \text{Energy}_{\text{nominal voltage}} - \left( \text{MACs}_{\text{base model}} \cdot \text{Energy}_{\text{low-voltage-regime}} + \text{MACs}_{\text{NeuralFuse}} \cdot \text{Energy}_{\text{NeuralFuse at nominal voltage}} \right)}{\text{MACs}_{\text{base model}} \cdot \text{Energy}_{\text{nominal voltage}}} \times 100\% \quad (6)$$

Table 8: The MACs-Based energy saving percentage (%) for different combinations of base models and NeuralFuse.

| Base Model | ConvL | ConvS | DeConvL | DeConvS | UNetL | UNetS |
| --- | --- | --- | --- | --- | --- | --- |
| ResNet18 | 16.2 | 28.7 | 19.0 | 26.6 | 23.2 | 28.7 |
| ResNet50 | 24.5 | 29.8 | 25.7 | 28.9 | 27.4 | 29.8 |
| VGG11 | -21.8 | 23.9 | -11.5 | 16.0 | 3.6 | 23.7 |
| VGG16 | 5.0 | 27.3 | 10.0 | 23.5 | 17.4 | 27.2 |
| VGG19 | 10.4 | 28.0 | 14.4 | 25.0 | 20.2 | 28.0 |

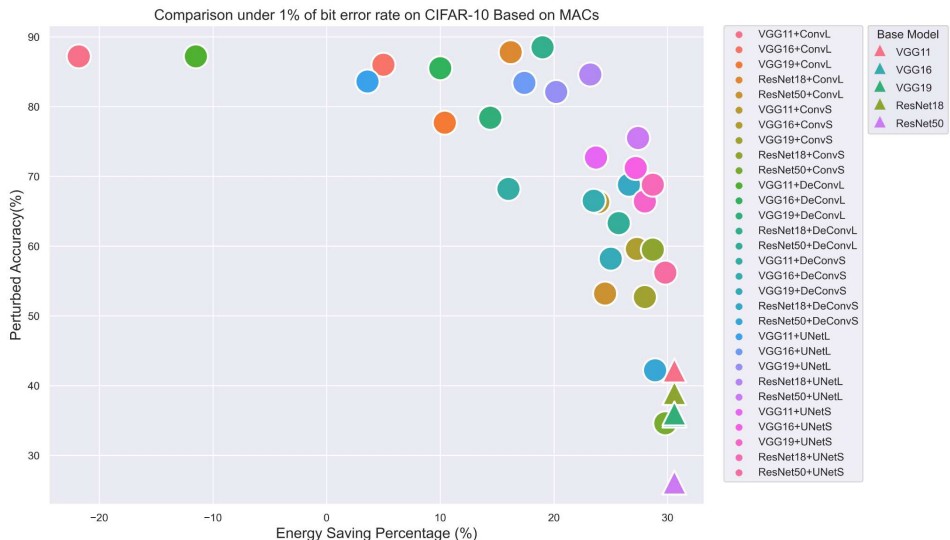

Figure 6: The MAC-Based energy-accuracy tradeoff of different NeuralFuse implementations with all CIFAR-10 pre-trained based models. X-axis represents the percentage reduction in dynamic memory access energy at low-voltage settings (base model protected by NeuralFuse), compared to the bit-error-free (nominal) voltage. Y-axis represents the perturbed accuracy (evaluated at low voltage) with a 1% bit error rate.

# F  ABLATION STUDIES

**Study for $N$ in EOPM.**  Here, we study the effect of $N$ used in EOPM (Eq. 5). In Figure 7, we report the results for ConvL and ConvS on CIFAR-10 pre-trained ResNet18, under a 1% bit error rate (B.E.R.). The results demonstrate that if we apply larger $N$, the performance increases until convergence. Specifically, for ConvL (Figure 7a), larger $N$ empirically has a smaller standard deviation; this means larger $N$ gives better stability but at the cost of time-consuming training. In contrast, for the small generator ConvS (Figure 7b), we can find that the standard deviation is still large even trained by larger $N$; the reason might be that small generators are not as capable of learning representations as larger ones. Therefore, there exists a trade-off between the *stability* of the generator performance and the total training time. In our implementation, choosing $N = 5$ or 10 is a good balance.

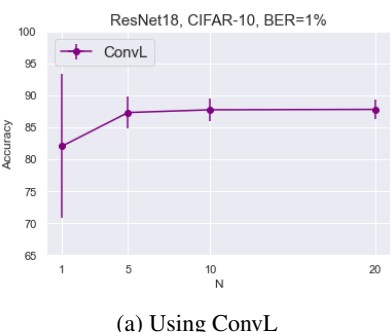
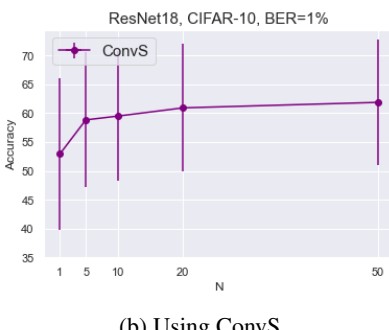

(a) Using ConvL  (b) Using ConvS

Figure 7: The experimental results on using different sizes of $N$ for EOPM.

**Tradeoff Between Clean Accuracy (C.A.) and Perturbed Accuracy (P.A.).**  We conducted an experiment to study the effect of different $\lambda$ values, which balance the ratio of clean accuracy and perturbed accuracy. In Table 9, the experimental results showed that a smaller $\lambda$ can preserve clean accuracy, but result in poor perturbed accuracy. On the contrary, larger $\lambda$ can deliver higher perturbed accuracy, but with more clean accuracy drop. This phenomenon has also been observed in adversarial training (Zhang et al., 2019).

Table 9: Experimental results based on $\lambda$ value choosing. The results show that $\lambda = 5$ can balance the tradeoff between clean accuracy and perturbed accuracy.

| Base Model | $\lambda$ | C.A. | P.A. | ConvL C.A. (NF) | ConvL P.A. (NF) | R.P. |
|---|---|---|---|---|---|---|
| ResNet18 | 10 | | | 90.1 | $88.0 \pm 1.7$ | 49.1 |
| | 5 | | | 89.8 | $87.8 \pm 1.7$ | 48.8 |
| | 1 | 92.6 | $38.9 \pm 12.4$ | 90.0 | $84.2 \pm 3.8$ | 45.3 |
| | 0.1 | | | 91.6 | $65.7 \pm 9.3$ | 26.8 |
| | 0.01 | | | 92.2 | $43.6 \pm 13$ | 4.7 |
| VGG19 | 10 | | | 89.6 | $77.9 \pm 19$ | 41.9 |
| | 5 | | | 89.8 | $77.7 \pm 19$ | 41.7 |
| | 1 | 90.5 | $36.0 \pm 12.0$ | 89.9 | $73.1 \pm 19$ | 37.1 |
| | 0.1 | | | 89.1 | $51.2 \pm 16$ | 15.2 |
| | 0.01 | | | 90.2 | $36.8 \pm 12$ | 0.8 |

[Note] C.A. (%): *clean accuracy*, P.A. (%): *perturbed accuracy*, NF: *NeuralFuse*, and R.P.: *total recover percentage of P.A. (NF) v.s. P.A.*

**Comparison to Universal Input Perturbation (UIP).**  Moosavi-Dezfooli et al. (2017) has shown that there exists a universal adversarial perturbation to the input data such that the model will make

wrong predictions on a majority of the perturbed images. In our NeuralFuse framework, the universal perturbation is a special case when we set $\mathcal{G}(\mathbf{x}) = \tanh(\mathbf{UIP})$ for any data sample $\mathbf{x}$. The transformed data sample then becomes $\mathbf{x}_t = \text{clip}_{[-1,1]}\big(\mathbf{x} + \tanh(\mathbf{UIP})\big)$, where $\mathbf{x}_t \in [-1,1]^d$ and **UIP** is a trainable universal input perturbation that has the same size as the input data. The experimental results with the universal input perturbation are shown in Table 10. We observe that its performance is much worse than our proposed NeuralFuse. This result validates the necessity of adopting input-aware transformation for learning error-resistant data representations in low-voltage scenarios.

Table 10: Performance of the universal input perturbation (UIP) trained by EOPM on CIFAR-10 pre-trained ResNet18.

| Base Model | B.E.R. | C.A. | P.A. | C.A. (**UIP**) | P.A. (**UIP**) | R.P. |
|---|---|---|---|---|---|---|
| ResNet18 | 1% | 92.6 | $38.9 \pm 12.4$ | 91.8 | $37.9 \pm 11$ | -1.0 |
|  | 0.5% |  | $70.1 \pm 11.6$ | 92.5 | $70.6 \pm 11$ | 0.5 |
| ResNet50 | 1% | 92.6 | $26.1 \pm 9.4$ | 80.7 | $21.0 \pm 5.9$ | -5.1 |
|  | 0.5% |  | $61.0 \pm 10.3$ | 91.9 | $62.4 \pm 12$ | 1.4 |
| VGG11 | 1% | 88.4 | $42.2 \pm 11.6$ | 86.9 | $43.0 \pm 11$ | 0.8 |
|  | 0.5% |  | $63.6 \pm 9.3$ | 88.2 | $64.2 \pm 8.8$ | 0.6 |
| VGG16 | 1% | 90.3 | $35.7 \pm 7.9$ | 90.1 | $37.1 \pm 8.5$ | 1.4 |
|  | 0.5% |  | $66.6 \pm 8.1$ | 90.4 | $67.3 \pm 8.1$ | 0.7 |
| VGG19 | 1% | 90.5 | $36.0 \pm 12.0$ | 89.9 | $35.3 \pm 12$ | -0.7 |
|  | 0.5% |  | $64.2 \pm 12.4$ | 90.1 | $64.4 \pm 12$ | 0.2 |

[Note] B.E.R.: *the bit error rate of the base model*, C.A. (%): *clean accuracy*, **UIP**: *universal input transformation parameter*, P.A.(%): *perturbed accuracy*, and R.P.: *total recover percentage of P.A. (UIP) v.s. P.A.*

## G  ADDITIONAL EXPERIMENTAL RESULTS ON RELAXED ACCESS SETTINGS

We conducted more experiments on *Relaxed Access* settings to show that our NeuralFuse can protect the models under different B.E.R. The results can be found in Sec. G.1 (CIFAR-10), Sec. G.2 (GTSRB), Sec. G.3 (ImageNet-10), and Sec. G.4 (CIFAR-100). For comparison, we also visualize the experimental results in the figures below each table.

### G.1  CIFAR-10

Table 11: Testing accuracy (%) under 1% and 0.5% of random bit error rate on CIFAR-10.

| Base Model | NF | C.A. | 1% B.E.R. | | | | 0.5% B.E.R. | | | |
|---|---|---|---|---|---|---|---|---|---|---|
| | | | P.A. | C.A. (NF) | P.A. (NF) | R.P. | P.A. | C.A. (NF) | P.A. (NF) | R.P. |
| ResNet18 | ConvL | 92.6 | 38.9 ± 12.4 | 89.8 | 87.8 ± 1.7 | 48.8 | 70.1 ± 11.6 | 90.4 | 87.9 ± 2.2 | 17.8 |
| | ConvS | | | 88.2 | 59.5 ± 11 | 20.6 | | 91.7 | 78.4 ± 8.3 | 8.3 |
| | DeConvL | | | 89.6 | 88.5 ± 0.8 | 49.6 | | 90.2 | 90.0 ± 0.2 | 19.9 |
| | DeConvS | | | 82.9 | 68.8 ± 6.4 | 29.9 | | 84.1 | 79.9 ± 3.6 | 9.8 |
| | UNetL | | | 86.6 | 84.6 ± 0.8 | 45.6 | | 89.7 | 86.3 ± 2.4 | 16.2 |
| | UNetS | | | 84.4 | 68.8 ± 6.0 | 29.8 | | 90.9 | 80.7 ± 5.8 | 10.7 |
| ResNet50 | ConvL | 92.6 | 26.1 ± 9.4 | 85.5 | 53.2 ± 22 | 27.1 | 61.0 ± 10.3 | 90.3 | 86.5 ± 3.2 | 25.5 |
| | ConvS | | | 85.2 | 34.6 ± 14 | 8.5 | | 90.8 | 73.3 ± 8.7 | 12.3 |
| | DeConvL | | | 87.4 | 63.3 ± 21 | 37.2 | | 89.5 | 87.2 ± 2.5 | 26.2 |
| | DeConvS | | | 82.4 | 42.2 ± 17 | 16.1 | | 90.3 | 75.5 ± 8.1 | 14.5 |
| | UNetL | | | 86.2 | 75.5 ± 12 | 49.4 | | 89.9 | 83.9 ± 3.6 | 22.9 |
| | UNetS | | | 77.3 | 56.2 ± 19 | 30.1 | | 89.7 | 76.1 ± 7.2 | 15.1 |
| VGG11 | ConvL | 88.4 | 42.2 ± 11.6 | 89.6 | 87.2 ± 2.9 | 45.1 | 63.6 ± 9.3 | 89.8 | 87.0 ± 1.3 | 23.3 |
| | ConvS | | | 84.9 | 66.3 ± 7.5 | 24.1 | | 88.2 | 74.5 ± 5.7 | 10.9 |
| | DeConvL | | | 89.3 | 87.2 ± 2.6 | 45.0 | | 89.6 | 86.9 ± 1.1 | 23.2 |
| | DeConvS | | | 85.6 | 68.2 ± 7.1 | 26.0 | | 88.3 | 75.7 ± 4.6 | 12.1 |
| | UNetL | | | 87.1 | 83.6 ± 1.3 | 41.4 | | 88.0 | 82.4 ± 1.8 | 18.8 |
| | UNetS | | | 85.5 | 72.7 ± 4.6 | 30.5 | | 88.1 | 75.8 ± 4.3 | 12.2 |
| VGG16 | ConvL | 90.3 | 35.7 ± 7.9 | 90.1 | 86.0 ± 6.2 | 50.3 | 66.6 ± 8.1 | 90.2 | 88.5 ± 0.9 | 21.9 |
| | ConvS | | | 87.4 | 59.6 ± 12 | 23.9 | | 89.9 | 77.8 ± 4.8 | 11.1 |
| | DeConvL | | | 89.7 | 85.5 ± 6.8 | 49.8 | | 89.7 | 88.2 ± 1.0 | 21.4 |
| | DeConvS | | | 86.8 | 66.5 ± 11 | 30.8 | | 90.0 | 78.4 ± 4.7 | 11.8 |
| | UNetL | | | 87.4 | 83.4 ± 4.4 | 47.7 | | 89.0 | 86.2 ± 1.5 | 19.6 |
| | UNetS | | | 87.4 | 71.2 ± 8.2 | 35.5 | | 89.0 | 80.2 ± 3.5 | 13.7 |
| VGG19 | ConvL | 90.5 | 36.0 ± 12.0 | 89.8 | 77.7 ± 19 | 41.7 | 64.2 ± 12.4 | 90.4 | 88.1 ± 1.8 | 23.9 |
| | ConvS | | | 87.3 | 52.7 ± 17 | 16.7 | | 89.6 | 74.5 ± 9.0 | 10.3 |
| | DeConvL | | | 86.3 | 78.4 ± 18 | 42.4 | | 90.4 | 88.5 ± 1.4 | 24.3 |
| | DeConvS | | | 86.5 | 58.2 ± 18 | 22.2 | | 89.7 | 75.2 ± 8.6 | 11.0 |
| | UNetL | | | 86.3 | 82.1 ± 4.8 | 46.0 | | 89.1 | 85.0 ± 2.7 | 20.8 |
| | UNetS | | | 86.3 | 66.4 ± 13 | 30.4 | | 89.2 | 77.1 ± 7.3 | 12.9 |

[Note] C.A. (%): *clean accuracy*, P.A. (%): *perturbed accuracy*, NF: *NeuralFuse*, and R.P.: *total recover percentage of P.A. (NF) v.s. P.A.*

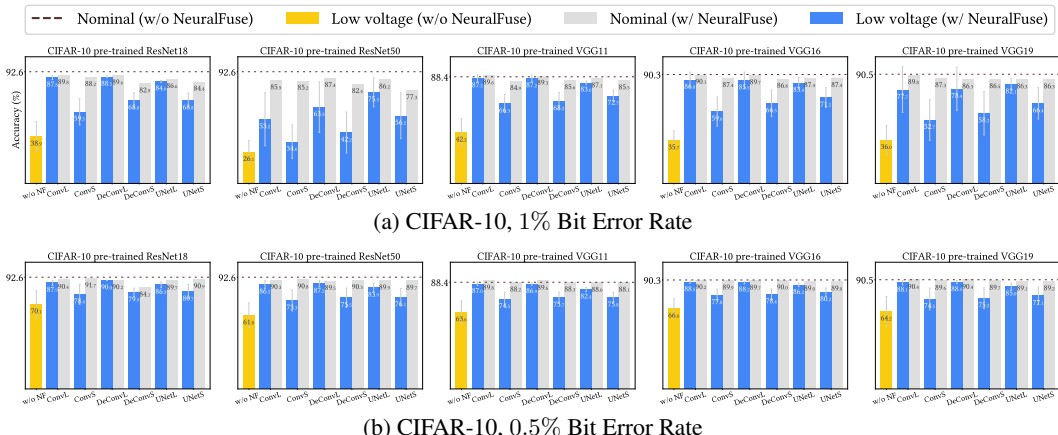

(a) CIFAR-10, 1% Bit Error Rate

(b) CIFAR-10, 0.5% Bit Error Rate

Figure 8: Experimental results on CIFAR-10

## G.2 GTSRB

Table 12: Testing accuracy (%) under 1% and 0.5% of random bit error rate on GTSRB.

| Base Model | NF | C.A. | 1% B.E.R. | | | | 0.5% B.E.R. | | | |
|---|---|---|---|---|---|---|---|---|---|---|
| | | | P.A. | C.A. (NF) | P.A. (NF) | R.P. | P.A. | C.A. (NF) | P.A. (NF) | R.P. |
| ResNet18 | ConvL | 95.5 | 36.9 ± 16.0 | 95.7 | 91.1 ± 4.7 | 54.2 | 75.2 ± 12.7 | 93.4 | 89.5 ± 1.9 | 14.3 |
| | ConvS | | | 94.4 | 68.6 ± 12 | 31.7 | | 94.8 | 87.7 ± 4.2 | 12.4 |
| | DeConvL | | | 95.6 | 91.3 ± 4.3 | 54.4 | | 95.4 | 93.4 ± 1.1 | 18.1 |
| | DeConvS | | | 95.7 | 78.1 ± 9.1 | 41.2 | | 95.8 | 90.1 ± 3.3 | 14.9 |
| | UNetL | | | 96.2 | 93.8 ± 1.0 | 56.9 | | 96.2 | 93.5 ± 1.6 | 18.3 |
| | UNetS | | | 95.9 | 85.1 ± 6.9 | 48.2 | | 95.5 | 91.4 ± 2.8 | 16.2 |
| ResNet50 | ConvL | 95.0 | 29.5 ± 16.9 | 95.6 | 71.6 ± 20 | 42.1 | 74.0 ± 13.0 | 94.6 | 90.6 ± 3.7 | 16.6 |
| | ConvS | | | 94.8 | 50.5 ± 22 | 21.0 | | 95.4 | 84.5 ± 8.5 | 10.5 |
| | DeConvL | | | 94.9 | 71.6 ± 21 | 42.0 | | 94.7 | 91.6 ± 2.9 | 17.6 |
| | DeConvS | | | 93.0 | 56.4 ± 17 | 26.9 | | 94.6 | 87.4 ± 5.9 | 13.5 |
| | UNetL | | | 94.5 | 80.6 ± 15 | 51.1 | | 96.5 | 93.7 ± 2.3 | 19.7 |
| | UNetS | | | 94.7 | 64.7 ± 22 | 35.2 | | 95.9 | 90.6 ± 4.8 | 16.7 |
| VGG11 | ConvL | 91.9 | 34.9 ± 12.4 | 94.8 | 85.7 ± 7.2 | 50.9 | 64.9 ± 10.8 | 93.9 | 92.6 ± 0.7 | 27.7 |
| | ConvS | | | 91.1 | 62.2 ± 11 | 27.3 | | 90.9 | 80.5 ± 3.5 | 15.7 |
| | DeConvL | | | 95.0 | 84.6 ± 7.6 | 49.7 | | 93.6 | 91.9 ± 0.6 | 27.1 |
| | DeConvS | | | 92.4 | 67.5 ± 11 | 32.6 | | 92.3 | 83.1 ± 3.7 | 18.2 |
| | UNetL | | | 92.2 | 83.2 ± 6.0 | 48.3 | | 94.8 | 90.6 ± 1.7 | 25.7 |
| | UNetS | | | 94.7 | 73.4 ± 10 | 38.5 | | 94.6 | 88.9 ± 2.2 | 24.1 |
| VGG16 | ConvL | 95.2 | 15.1 ± 6.8 | 96.3 | 72.4 ± 12 | 57.3 | 58.8 ± 8.9 | 95.6 | 93.2 ± 1.8 | 34.4 |
| | ConvS | | | 94.1 | 39.8 ± 13 | 24.6 | | 94.3 | 82.2 ± 6.2 | 23.4 |
| | DeConvL | | | 96.4 | 72.0 ± 12 | 56.9 | | 95.6 | 93.1 ± 2.0 | 34.3 |
| | DeConvS | | | 93.8 | 50.9 ± 13 | 35.8 | | 95.1 | 84.0 ± 5.3 | 25.2 |
| | UNetL | | | 95.8 | 78.6 ± 11 | 63.5 | | 96.0 | 92.8 ± 2.0 | 34.0 |
| | UNetS | | | 94.3 | 63.3 ± 14 | 48.1 | | 95.4 | 87.8 ± 3.6 | 29.0 |
| VGG19 | ConvL | 95.5 | 36.6 ± 6.8 | 96.0 | 88.3 ± 7.2 | 51.7 | 69.1 ± 11.1 | 95.6 | 93.4 ± 2.1 | 24.2 |
| | ConvS | | | 93.8 | 69.0 ± 14 | 32.4 | | 94.9 | 87.0 ± 4.4 | 17.8 |
| | DeConvL | | | 95.4 | 87.2 ± 7.5 | 50.6 | | 95.5 | 92.4 ± 2.2 | 23.3 |
| | DeConvS | | | 94.5 | 73.1 ± 12 | 36.5 | | 95.5 | 88.8 ± 3.7 | 19.7 |
| | UNetL | | | 95.4 | 88.2 ± 6.7 | 51.7 | | 94.9 | 91.7 ± 2.5 | 22.6 |
| | UNetS | | | 94.6 | 80.6 ± 9.0 | 44.1 | | 96.5 | 90.8 ± 3.4 | 21.6 |

[Note] C.A. (%): *clean accuracy*, P.A. (%): *perturbed accuracy*, NF: *NeuralFuse*, and R.P.: *total recover percentage of P.A. (NF) v.s. P.A.*

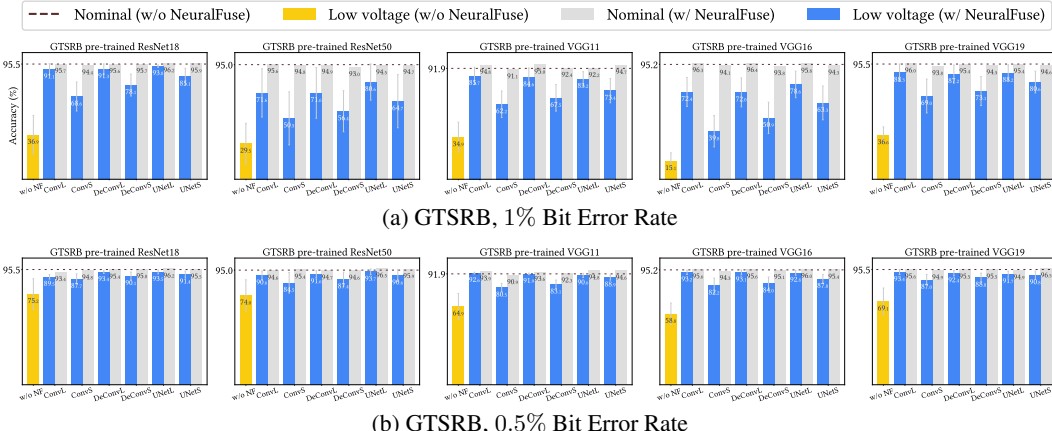

(a) GTSRB, 1% Bit Error Rate

(b) GTSRB, 0.5% Bit Error Rate

Figure 9: Experimental results on GTSRB.

### G.3   IMAGENET-10

Table 13: Testing accuracy under 0.5% of random bit error rate on ImageNet-10.

| Base Model | NF | C.A. | 0.5% B.E.R. | | | |
|---|---|---|---|---|---|---|
| | | | P.A. | C.A. (NF) | P.A. (NF) | R.P. |
| ResNet18 | ConvL | 92.2 | 72.3 ± 7.0 | 94.0 | 88.0 ± 2.0 | 15.7 |
| | ConvS | | | 91.8 | 83.6 ± 4.1 | 11.3 |
| | DeConvL | | | 94.0 | 89.2 ± 1.3 | 16.9 |
| | DeConvS | | | 92.8 | 87.5 ± 2.3 | 15.2 |
| | UNetL | | | 94.0 | 88.1 ± 1.4 | 15.8 |
| | UNetS | | | 93.2 | 86.4 ± 2.2 | 14.1 |
| ResNet50 | ConvL | 89.8 | 39.4 ± 11 | 92.2 | 80.0 ± 5.8 | 40.6 |
| | ConvS | | | 91.8 | 65.0 ± 11 | 25.6 |
| | DeConvL | | | 93.0 | 79.4 ± 5.9 | 40.0 |
| | DeConvS | | | 93.2 | 70.9 ± 9.1 | 31.5 |
| | UNetL | | | 92.2 | 80.5 ± 5.8 | 41.1 |
| | UNetS | | | 92.4 | 73.6 ± 8.9 | 34.2 |
| VGG11 | ConvL | 91.6 | 47.8 ± 13 | 92.0 | 86.1 ± 3.7 | 38.3 |
| | ConvS | | | 89.4 | 66.4 ± 7.1 | 18.6 |
| | DeConvL | | | 91.0 | 86.0 ± 3.0 | 38.2 |
| | DeConvS | | | 89.0 | 72.5 ± 7.8 | 24.7 |
| | UNetL | | | 92.4 | 83.0 ± 3.5 | 35.2 |
| | UNetS | | | 86.2 | 73.5 ± 6.0 | 25.7 |
| VGG16 | ConvL | 94.6 | 38.4 ± 17 | 90.8 | 77.1 ± 11 | 38.7 |
| | ConvS | | | 90.2 | 60.2 ± 14 | 21.8 |
| | DeConvL | | | 91.2 | 77.2 ± 11 | 38.8 |
| | DeConvS | | | 90.0 | 62.3 ± 14 | 23.9 |
| | UNetL | | | 90.6 | 81.1 ± 5.9 | 42.7 |
| | UNetS | | | 86.4 | 72.3 ± 8.8 | 33.9 |
| VGG19 | ConvL | 92.4 | 37.2 ± 11 | 91.4 | 75.5 ± 8.8 | 38.3 |
| | ConvS | | | 88.8 | 56.5 ± 13 | 19.3 |
| | DeConvL | | | 91.0 | 75.9 ± 8.9 | 38.7 |
| | DeConvS | | | 88.8 | 64.0 ± 11 | 26.8 |
| | UNetL | | | 89.4 | 77.9 ± 6.1 | 40.7 |
| | UNetS | | | 87.6 | 65.9 ± 10 | 28.7 |

[Note] C.A. (%): *clean accuracy*, P.A. (%): *perturbed accuracy*, NF: *NeuralFuse*, and R.P.: *total recover percentage of P.A. (NF) v.s. P.A.*

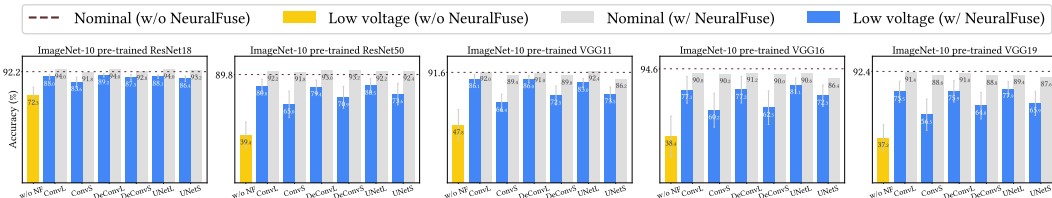

Figure 10: Experimental results on ImageNet-10, 0.5% Bit Error Rate.

## G.4 CIFAR-100

As mentioned in the previous section, larger generators like ConvL, DeConvL, and UNetL have better performance than small generators. For CIFAR-100, we find that the gains of utilizing NeuralFuse are less compared to the other datasets. We believe this is because CIFAR-100 is a more challenging dataset (more classes) for the generators to learn to protect the base models. Nevertheless, NeuralFuse can still function to restore some degraded accuracy; these results also demonstrate that our NeuralFuse is applicable to different datasets. In addition, although the recover percentage is less obvious on CIFAR-100 (the more difficult dataset), we can still conclude that our NeuralFuse is applicable to different datasets.

Table 14: Testing accuracy (%) under 1%, 0.5% and 0.35% of random bit error rate on CIFAR-100.

| Base Model | NF | C.A. | 1% B.E.R. | | | | | 0.5% B.E.R. | | | | | 0.35% B.E.R. | | | |
|---|---|---|---|---|---|---|---|---|---|---|---|---|---|---|---|---|
| | | | P.A. | C.A. (NF) | P.A. (NF) | R.P. | P.A. | C.A. (NF) | P.A. (NF) | R.P. | P.A. | C.A. (NF) | P.A. (NF) | R.P. | | |
| ResNet18 | ConvL | 73.7 ± 2.9 | 4.6 ± 2.9 | 54.8 | 11.0 ± 7.7 | 6.4 | 20.9 ± 7.4 | 65.2 | 39.0 ± 7.1 | 18.1 | 31.4 ± 7.6 | 69.4 | 42.9 ± 6.2 | 11.4 | | |
| | ConvS | | | 49.7 | 4.2 ± 2.2 | -0.4 | | 70.0 | 24.5 ± 7.6 | 3.6 | | 72.1 | 35.1 ± 7.3 | 3.7 | | |
| | DeConvL | | | 55.2 | 11.9 ± 8.2 | 7.3 | | 66.3 | 38.2 ± 6.9 | 17.3 | | 69.2 | 42.9 ± 5.5 | 11.4 | | |
| | DeConvS | | | 32.7 | 4.0 ± 2.2 | -0.6 | | 68.2 | 25.9 ± 6.8 | 5 | | 71.6 | 35.8 ± 5.5 | 4.4 | | |
| | UNetL | | | 50.6 | 14.5 ± 8.9 | 10.0 | | 66.2 | 40.1 ± 6.4 | 19.2 | | 70.3 | 46.3 ± 5.5 | 14.9 | | |
| | UNetS | | | 26.8 | 4.6 ± 2.5 | -0.0 | | 67.1 | 28.8 ± 6.8 | 7.9 | | 70.9 | 38.3 ± 6.4 | 6.9 | | |
| ResNet50 | ConvL | 73.5 ± 1.8 | 3.0 ± 1.8 | 63.5 | 3.2 ± 1.7 | 0.1 | 21.3 ± 7.0 | 68.4 | 28.8 ± 6.7 | 7.6 | 35.7 ± 8.6 | 72.0 | 40.8 ± 7.5 | 5.1 | | |
| | ConvS | | | 65.5 | 3.2 ± 1.6 | 0.1 | | 71.9 | 23.1 ± 6.9 | 1.9 | | 73.0 | 37.4 ± 8.0 | 1.7 | | |
| | DeConvL | | | 59.6 | 3.2 ± 1.7 | 0.2 | | 68.1 | 28.6 ± 7.0 | 7.4 | | 71.7 | 41.7 ± 7.7 | 6.1 | | |
| | DeConvS | | | 61.1 | 3.2 ± 1.7 | 0.1 | | 70.3 | 25.0 ± 6.7 | 3.7 | | 72.8 | 38.9 ± 7.9 | 3.3 | | |
| | UNetL | | | 39.0 | 5.0 ± 1.7 | 1.9 | | 66.6 | 36.5 ± 6.2 | 15.3 | | 70.8 | 45.3 ± 6.7 | 9.6 | | |
| | UNetS | | | 47.7 | 3.4 ± 1.8 | 0.3 | | 69.1 | 26.1 ± 6.6 | 4.8 | | 72.6 | 39.6 ± 7.8 | 3.9 | | |
| VGG11 | ConvL | 64.8 ± 5.7 | 8.2 ± 5.7 | 58.3 | 19.7 ± 11 | 11.5 | 23.9 ± 9.4 | 63.1 | 38.8 ± 9.3 | 15.0 | 31.3 ± 10 | 63.9 | 42.4 ± 9.0 | 11.1 | | |
| | ConvS | | | 56.6 | 10.4 ± 7.4 | 2.2 | | 62.7 | 27.9 ± 10 | 4.0 | | 63.9 | 41.8 ± 8.3 | 10.5 | | |
| | DeConvL | | | 60.3 | 21.2 ± 11 | 13.0 | | 63.9 | 40.0 ± 9.0 | 16.2 | | 64.0 | 42.8 ± 9.1 | 11.5 | | |
| | DeConvS | | | 58.3 | 11.8 ± 7.9 | 3.5 | | 61.9 | 29.8 ± 9.9 | 5.9 | | 63.5 | 36.1 ± 10 | 4.8 | | |
| | UNetL | | | 51.1 | 22.1 ± 8.2 | 13.9 | | 61.8 | 37.8 ± 9.0 | 13.9 | | 63.5 | 40.9 ± 9.3 | 9.6 | | |
| | UNetS | | | 51.9 | 13.1 ± 7.9 | 4.9 | | 61.7 | 29.8 ± 9.7 | 6.0 | | 63.8 | 35.7 ± 9.9 | 4.5 | | |
| VGG16 | ConvL | 67.8 ± 3.5 | 7.0 ± 3.5 | 51.4 | 19.2 ± 6.0 | 12.6 | 22.4 ± 7.0 | 61.8 | 41.1 ± 5.6 | 18.7 | 31.1 ± 7.2 | 64.9 | 44.9 ± 5.3 | 13.8 | | |
| | ConvS | | | 44.3 | 6.7 ± 2.3 | 0.1 | | 63.8 | 27.5 ± 6.8 | 5.1 | | 66.0 | 36.3 ± 6.1 | 5.1 | | |
| | DeConvL | | | 53.1 | 20.8 ± 6.2 | 14.2 | | 62.8 | 42.1 ± 5.5 | 19.8 | | 65.0 | 46.6 ± 5.2 | 15.5 | | |
| | DeConvS | | | 23.5 | 4.8 ± 1.7 | -1.8 | | 62.1 | 29.9 ± 6.7 | 7.5 | | 64.9 | 38.1 ± 6.3 | 7.0 | | |
| | UNetL | | | 50.2 | 25.3 ± 1.7 | 18.7 | | 61.7 | 41.3 ± 5.0 | 18.9 | | 64.8 | 46.8 ± 4.6 | 15.7 | | |
| | UNetS | | | 27.7 | 9.9 ± 2.1 | 3.3 | | 61.6 | 31.3 ± 6.3 | 8.9 | | 65.0 | 39.8 ± 5.9 | 8.7 | | |
| VGG19 | ConvL | 67.8 ± 4.3 | 10.6 ± 4.3 | 59.4 | 29.2 ± 8.1 | 18.6 | 34.0 ± 9.6 | 65.6 | 46.5 ± 6.8 | 12.5 | 42.1 ± 9.4 | 66.9 | 49.2 ± 7.4 | 7.0 | | |
| | ConvS | | | 63.7 | 14.4 ± 5.1 | 3.8 | | 66.6 | 38.3 ± 6.8 | 4.2 | | 67.7 | 45.3 ± 8.5 | 3.2 | | |
| | DeConvL | | | 60.1 | 29.6 ± 8.5 | 19.0 | | 65.7 | 46.9 ± 7.1 | 12.9 | | 67.3 | 49.8 ± 7.6 | 7.6 | | |
| | DeConvS | | | 60.9 | 16.1 ± 6.0 | 5.6 | | 66.5 | 39.0 ± 3.7 | 5.0 | | 67.7 | 45.7 ± 8.4 | 3.6 | | |
| | UNetL | | | 58.7 | 30.2 ± 8.2 | 19.6 | | 65.5 | 46.9 ± 6.5 | 12.9 | | 67.4 | 50.0 ± 7.5 | 7.9 | | |
| | UNetS | | | 59.1 | 18.0 ± 6.2 | 7.4 | | 66.3 | 40.1 ± 8.0 | 6.1 | | 67.5 | 46.6 ± 8.4 | 4.5 | | |

[Note] C.A. (%): *clean accuracy*, P.A. (%): *perturbed accuracy*, NF: *NeuralFuse*, and R.P.: *total recover percentage of P.A. (NF) v.s. P.A.*

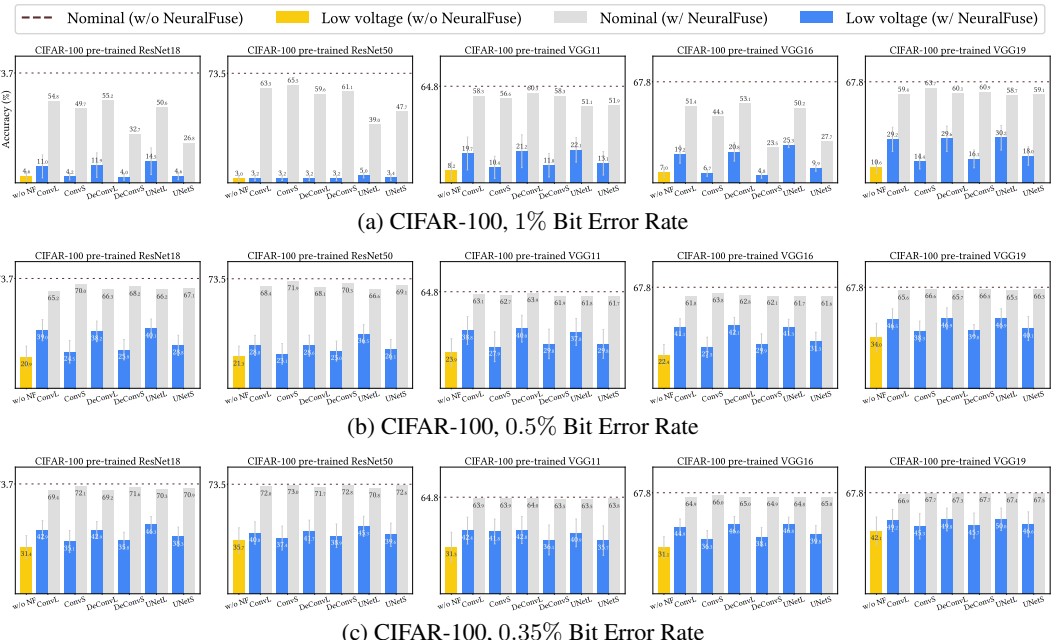

(a) CIFAR-100, 1% Bit Error Rate

(b) CIFAR-100, 0.5% Bit Error Rate

(c) CIFAR-100, 0.35% Bit Error Rate

Figure 11: Experimental results on CIFAR-100.

# H ADDITIONAL EXPERIMENTAL RESULTS ON RESTRICTED ACCESS SETTINGS (TRANSFERABILITY)

We conduct more experiments with *Restricted Access* settings to show that our NeuralFuse can be transferred to protect various black-box models. The experimental results are shown in Sec. H.1 (CIFAR-10), Sec. H.2 (GTSRB), and Sec. H.3 (CIFAR-100).

We find that using VGG19 as a white-box surrogate model has better *transferability* than ResNet18 for all datasets. In addition, we can observe that some NeuralFuse generators have *downward applicability* if base models have a similar architecture. In other words, if we try to transfer a generator trained on a large B.E.R. (e.g., 1%) to a model with a small B.E.R. (e.g., 0.5%), the performance will be better than that of a generator trained with the original B.E.R. (e.g., 0.5%). For example, in Table 15, we could find that if we use VGG19 as a source model to train the generator ConvL (1%), the generator could deliver better performance (in terms of P.A. (NF)) when applied to similar base models (e.g., VGG11, VGG16, or VGG19) under a 0.5% B.E.R., compared to using itself as a source model (shown in Table 11). We conjecture that this is because the generators trained on a larger B.E.R. can also cover the error patterns of a smaller B.E.R., and thus they have better generalizability across smaller B.E.Rs.

To further improve the transferability to cross-architecture target models, we also conduct an experiment in Sec. H.4 to show that using ensemble-based training can help the generator to achieve this feature.

## H.1 CIFAR-10

The results of CIFAR-10 in which NeuralFuse is trained at 1% B.E.R. are shown in Table 15.

Table 15: Transfer results on CIFAR-10: NeuralFuse trained on S.M. with 1% B.E.R.

| S.M. | T.M. | B.E.R. | C.A. | P.A. | ConvL (1%) | | | UNetL (1%) | | |
|------|------|--------|------|------|------------|------|------|------------|------|------|
| | | | | | C.A. (NF) | P.A. (NF) | R.P. | C.A. (NF) | P.A. (NF) | R.P. |
| ResNet18 | ResNet18 | 0.5% | 92.6 | 70.1 ± 11.6 | 89.8 | 89.5 ± 0.2 | 19.4 | 86.6 | 86.2 ± 0.3 | 16.1 |
| | ResNet50 | 1% | 92.6 | 26.1 ± 9.4 | 89.5 | 36.0 ± 19 | 9.9 | 85.2 | 38.8 ± 19 | 12.7 |
| | | 0.5% | | 61.0 ± 10.3 | | 75.1 ± 10 | 14.1 | | 77.1 ± 5.0 | 16.1 |
| | VGG11 | 1% | 88.4 | 42.2 ± 11.6 | 88.4 | 62.5 ± 8.4 | 20.3 | 76.8 | 61.1 ± 8.5 | 18.9 |
| | | 0.5% | | 63.6 ± 9.3 | | 81.0 ± 4.6 | 17.4 | | 73.7 ± 3.0 | 10.1 |
| | VGG16 | 1% | 90.3 | 35.7 ± 7.9 | 89.6 | 63.3 ± 18 | 27.6 | 85.2 | 59.9 ± 16 | 24.2 |
| | | 0.5% | | 66.6 ± 8.1 | | 85.0 ± 3.4 | 18.4 | | 80.2 ± 4.5 | 13.6 |
| | VGG19 | 1% | 90.5 | 36.0 ± 12.0 | 89.6 | 50.7 ± 22 | 14.7 | 85.3 | 51.1 ± 16 | 15.1 |
| | | 0.5% | | 64.2 ± 12.4 | | 80.2 ± 8.7 | 16.0 | | 76.5 ± 7.8 | 12.3 |
| VGG19 | ResNet18 | 1% | 92.6 | 38.9 ± 12.4 | 89.8 | 61.0 ± 17 | 22.1 | 87.0 | 69.7 ± 11 | 30.8 |
| | | 0.5% | | 70.1 ± 11.6 | | 86.1 ± 6.9 | 16.0 | | 84.2 ± 3.0 | 14.1 |
| | ResNet50 | 1% | 92.6 | 26.1 ± 9.4 | 89.9 | 34.0 ± 19 | 7.9 | 87.0 | 44.2 ± 17 | 18.1 |
| | | 0.5% | | 61.0 ± 10.3 | | 76.5 ± 10 | 15.5 | | 80.7 ± 4.2 | 19.7 |
| | VGG11 | 1% | 88.4 | 42.2 ± 11.6 | 89.7 | 76.5 ± 7.0 | 34.3 | 87.1 | 79.9 ± 5.6 | 37.7 |
| | | 0.5% | | 63.6 ± 9.3 | | 88.0 ± 2.1 | 24.4 | | 85.4 ± 0.8 | 21.8 |
| | VGG16 | 1% | 90.3 | 35.7 ± 7.9 | 89.6 | 75.5 ± 12 | 39.8 | 87.2 | 78.9 ± 7.8 | 43.2 |
| | | 0.5% | | 66.6 ± 8.1 | | 88.9 ± 0.6 | 22.3 | | 86.2 ± 0.3 | 19.6 |
| | VGG19 | 0.5% | 90.5 | 64.2 ± 12.4 | 89.8 | 89.6 ± 8.7 | 25.4 | 87.4 | 86.8 ± 0.4 | 22.6 |

[Note] S.M.: *source model, used for training generators*, T.M.: *target model, used for testing generators*, B.E.R.: *the bit error rate of the target model*, C.A. (%): *clean accuracy*, P.A. (%): *perturbed accuracy*, NF: *NeuralFuse*, and R.P.: *total recover percentage of P.A. (NF) v.s. P.A.*

## H.2 GTSRB

In Tables 16 and 17, we show the results on GTSRB in which NeuralFuse is trained at 1.5% and 1% B.E.R., respectively.

Table 16: Transfer results on GTSRB: NeuralFuse trained on S.M. with 1.5% B.E.R.

| S.M. | T.M. | B.E.R. | C.A. | P.A. | ConvL (1.5%) | | | UNetL (1.5%) | | |
|---|---|---|---|---|---|---|---|---|---|---|
| | | | | | C.A. (NF) | P.A. (NF) | R.P. | C.A. (NF) | P.A. (NF) | R.P. |
| ResNet18 | ResNet18 | 1% | 95.5 | 36.9 ± 16.0 | 95.7 | 93.9 ± 1.9 | 57.0 | 94.9 | 94.4 ± 0.4 | 57.5 |
| | | 0.5% | | 75.2 ± 12.7 | | 95.7 ± 0.2 | 20.5 | | 94.8 ± 0.2 | 19.6 |
| | ResNet50 | 1% | 95.0 | 29.5 ± 16.9 | 94.4 | 37.0 ± 22 | 7.5 | 94.4 | 47.1 ± 23 | 17.6 |
| | | 0.5% | | 74.0 ± 13.0 | | 77.5 ± 13 | 3.5 | | 84.8 ± 9.5 | 10.8 |
| | VGG11 | 1% | 91.9 | 34.9 ± 12.4 | 92.8 | 45.2 ± 10 | 10.3 | 91.4 | 50.5 ± 13 | 15.6 |
| | | 0.5% | | 64.9 ± 10.8 | | 79.4 ± 5.8 | 14.5 | | 83.9 ± 4.2 | 19.0 |
| | VGG16 | 1% | 95.2 | 15.1 ± 6.8 | 95.4 | 31.1 ± 13 | 15.9 | 94.6 | 36.8 ± 12 | 21.7 |
| | | 0.5% | | 58.8 ± 8.9 | | 84.5 ± 8.3 | 25.8 | | 86.0 ± 8.6 | 27.2 |
| | VGG19 | 1% | 95.5 | 36.6 ± 6.8 | 95.0 | 56.4 ± 15 | 19.8 | 94.3 | 60.8 ± 15 | 24.2 |
| | | 0.5% | | 69.1 ± 11.1 | | 86.9 ± 3.4 | 17.8 | | 87.7 ± 3.8 | 18.6 |
| VGG19 | ResNet18 | 1% | 95.5 | 36.9 ± 16.0 | 88.4 | 50.3 ± 12 | 13.4 | 92.8 | 63.7 ± 16 | 26.8 |
| | | 0.5% | | 75.2 ± 12.7 | | 77.9 ± 7.4 | 2.7 | | 87.5 ± 3.9 | 12.3 |
| | ResNet50 | 1% | 95.0 | 29.5 ± 16.9 | 87.5 | 29.7 ± 17 | 0.2 | 92.5 | 40.4 ± 21 | 10.9 |
| | | 0.5% | | 74.0 ± 13.0 | | 67.9 ± 17 | -6.1 | | 77.5 ± 15 | 3.5 |
| | VGG11 | 1% | 91.9 | 34.9 ± 12.4 | 89.7 | 47.1 ± 11 | 12.2 | 93.5 | 60.0 ± 12 | 25.1 |
| | | 0.5% | | 64.9 ± 10.8 | | 76.3 ± 5.1 | 11.4 | | 86.0 ± 3.8 | 21.1 |
| | VGG16 | 1% | 95.2 | 15.1 ± 6.8 | 93.0 | 29.2 ± 15 | 14.1 | 93.0 | 38.5 ± 16 | 23.4 |
| | | 0.5% | | 58.8 ± 8.9 | | 75.7 ± 12 | 16.9 | | 79.9 ± 8.3 | 21.1 |
| | VGG19 | 1% | 95.5 | 36.6 ± 6.8 | 95.1 | 87.4 ± 6.0 | 50.8 | 94.6 | 88.7 ± 5.0 | 52.1 |
| | | 0.5% | | 69.1 ± 11.1 | | 92.4 ± 2.4 | 23.3 | | 92.4 ± 2.2 | 23.3 |

[Note] S.M.: *source model, used for training generators*, T.M.: *target model, used for testing generators*, B.E.R.: *the bit error rate of the target model*, C.A. (%): *clean accuracy*, P.A. (%): *perturbed accuracy*, NF: *NeuralFuse*, and R.P.: *total recover percentage of P.A. (NF) v.s. P.A.*

Table 17: Transfer results on GTSRB: NeuralFuse trained on S.M. with 1% B.E.R.

| S.M. | T.M. | B.E.R. | C.A. | P.A. | ConvL (1%) | | | UNetL (1%) | | |
|---|---|---|---|---|---|---|---|---|---|---|
| | | | | | C.A. (NF) | P.A. (NF) | R.P. | C.A. (NF) | P.A. (NF) | R.P. |
| ResNet18 | ResNet18 | 0.5% | 95.5 | 75.2 ± 12.7 | 95.7 | 95.3 ± 0.5 | 20.1 | 96.2 | 95.7 ± 0.3 | 20.5 |
| | ResNet50 | 1% | 95.0 | 29.5 ± 16.9 | 94.5 | 35.6 ± 21 | 6.1 | 95.6 | 42.6 ± 23 | 13.1 |
| | | 0.5% | | 74.0 ± 13.0 | | 78.8 ± 13 | 4.8 | | 87.3 ± 9.0 | 13.3 |
| | VGG11 | 1% | 91.9 | 34.9 ± 12.4 | 93.1 | 45.8 ± 11 | 10.9 | 94.0 | 47.1 ± 14 | 12.2 |
| | | 0.5% | | 64.9 ± 10.8 | | 81.8 ± 5.0 | 16.9 | | 84.2 ± 4.8 | 19.3 |
| | VGG16 | 1% | 95.2 | 15.1 ± 6.8 | 95.5 | 26.5 ± 12 | 11.4 | 95.5 | 32.4 ± 11 | 17.3 |
| | | 0.5% | | 58.8 ± 8.9 | | 82.2 ± 9.0 | 23.4 | | 85.4 ± 6.7 | 26.6 |
| | VGG19 | 1% | 95.5 | 36.6 ± 6.8 | 94.9 | 53.2 ± 14 | 16.6 | 95.6 | 60.9 ± 15 | 24.3 |
| | | 0.5% | | 69.1 ± 11.1 | | 85.4 ± 4.5 | 16.3 | | 87.5 ± 3.7 | 18.4 |
| VGG19 | ResNet18 | 1% | 95.5 | 36.9 ± 16.0 | 93.7 | 53.1 ± 16 | 16.2 | 95.0 | 63.4 ± 18 | 26.5 |
| | | 0.5% | | 75.2 ± 12.7 | | 83.9 ± 7.6 | 8.7 | | 89.7 ± 4.8 | 14.5 |
| | ResNet50 | 1% | 95.0 | 29.5 ± 16.9 | 92.8 | 30.6 ± 18 | 1.1 | 95.4 | 38.9 ± 22 | 9.4 |
| | | 0.5% | | 74.0 ± 13.0 | | 74.7 ± 18 | 0.7 | | 81.5 ± 16 | 7.5 |
| | VGG11 | 1% | 91.9 | 34.9 ± 12.4 | 93.7 | 50.6 ± 11 | 15.7 | 95.1 | 58.9 ± 15 | 24.0 |
| | | 0.5% | | 64.9 ± 10.8 | | 82.3 ± 5.1 | 17.4 | | 87.5 ± 3.7 | 22.6 |
| | VGG16 | 1% | 95.2 | 15.1 ± 6.8 | 95.2 | 27.8 ± 15 | 12.7 | 95.2 | 33.5 ± 14 | 18.4 |
| | | 0.5% | | 58.8 ± 8.9 | | 79.0 ± 12 | 20.2 | | 81.8 ± 7.8 | 23.0 |
| | VGG19 | 0.5% | 95.5 | 69.1 ± 11.1 | 96.0 | 94.0 ± 2.2 | 24.9 | 95.4 | 93.9 ± 2.1 | 24.8 |

[Note] S.M.: *source model, used for training generators*, T.M.: *target model, used for testing generators*, B.E.R.: *the bit error rate of the target model*, C.A. (%): *clean accuracy*, P.A. (%): *perturbed accuracy*, NF: *NeuralFuse*, and R.P.: *total recover percentage of P.A. (NF) v.s. P.A.*

## H.3   CIFAR-100

In Tables 18 and 19, we show results on CIFAR-100 with NeuralFuse trained at 1% and 0.5% B.E.R., respectively.

Table 18: Transfer results on CIFAR-100: NeuralFuse trained on S.M. with 1% B.E.R.

| S.M. | T.M. | B.E.R. | C.A. | P.A. | ConvL (1%) | | | UNetL (1%) | | |
|---|---|---|---|---|---|---|---|---|---|---|
| | | | | | C.A. (NF) | P.A. (NF) | R.P. | C.A. (NF) | P.A. (NF) | R.P. |
| ResNet18 | ResNet18 | 0.5% | 73.7 | $20.9 \pm 7.4$ | 54.8 | $35.8 \pm 5.2$ | 14.9 | 50.6 | $39.3 \pm 2.8$ | 18.4 |
| | | 0.35% | | $31.4 \pm 7.6$ | | $41.7 \pm 3.7$ | 10.3 | | $43.3 \pm 1.4$ | 11.9 |
| | ResNet50 | 1% | 73.5 | $3.0 \pm 1.8$ | 44.9 | $2.2 \pm 2.0$ | -0.8 | 41.5 | $2.4 \pm 1.9$ | -0.6 |
| | | 0.5% | | $21.3 \pm 7.0$ | | $15.9 \pm 8.2$ | -5.4 | | $17.1 \pm 7.1$ | -4.2 |
| | | 0.35% | | $35.7 \pm 8.6$ | | $23.7 \pm 7.1$ | -12.0 | | $26.2 \pm 5.6$ | -9.5 |
| | VGG11 | 1% | 64.8 | $8.2 \pm 5.7$ | 41.2 | $9.8 \pm 5.6$ | 1.6 | 37.5 | $10.2 \pm 5.1$ | 2.0 |
| | | 0.5% | | $23.9 \pm 9.4$ | | $24.2 \pm 5.9$ | 0.3 | | $24.5 \pm 4.7$ | 0.6 |
| | | 0.35% | | $31.3 \pm 10.0$ | | $29.0 \pm 5.4$ | -2.3 | | $28.2 \pm 4.5$ | -3.1 |
| | VGG16 | 1% | 67.8 | $7.0 \pm 3.5$ | 44.0 | $7.9 \pm 3.7$ | 0.9 | 39.5 | $10.1 \pm 4.5$ | 3.1 |
| | | 0.5% | | $22.4 \pm 7.0$ | | $22.4 \pm 7.6$ | 0.0 | | $26.3 \pm 5.3$ | 3.9 |
| | | 0.35% | | $31.1 \pm 7.2$ | | $28.1 \pm 5.9$ | -3.0 | | $30.6 \pm 3.6$ | -0.5 |
| | VGG19 | 1% | 67.8 | $10.6 \pm 4.3$ | 44.2 | $13.5 \pm 6.1$ | 2.9 | 40.7 | $15.6 \pm 6.2$ | 5.0 |
| | | 0.5% | | $34.0 \pm 9.6$ | | $27.9 \pm 4.8$ | -6.1 | | $29.3 \pm 4.6$ | -4.7 |
| | | 0.35% | | $42.1 \pm 9.4$ | | $33.2 \pm 48$ | -8.9 | | $32.8 \pm 3.9$ | -9.3 |
| VGG19 | ResNet18 | 1% | 73.7 | $4.6 \pm 2.9$ | 55.5 | $5.8 \pm 3.7$ | 1.2 | 57.3 | $6.8 \pm 4.4$ | 2.2 |
| | | 0.5% | | $20.9 \pm 7.4$ | | $24.6 \pm 6.3$ | 3.7 | | $28.1 \pm 5.9$ | 7.2 |
| | | 0.35% | | $31.4 \pm 7.6$ | | $31.1 \pm 5.0$ | -0.3 | | $36.4 \pm 4.5$ | 5.0 |
| | ResNet50 | 1% | 73.5 | $3.0 \pm 1.8$ | 56.1 | $2.8 \pm 2.1$ | -0.2 | 56.1 | $3.7 \pm 2.4$ | 0.7 |
| | | 0.5% | | $21.3 \pm 7.0$ | | $18.9 \pm 8.6$ | -2.4 | | $22.8 \pm 8.5$ | 1.5 |
| | | 0.35% | | $35.7 \pm 8.6$ | | $28.7 \pm 8.2$ | -7.0 | | $33.7 \pm 7.0$ | -2.0 |
| | VGG11 | 1% | 64.8 | $8.2 \pm 5.7$ | 52.8 | $12.3 \pm 8.4$ | 4.1 | 53.9 | $15.4 \pm 9.4$ | 7.2 |
| | | 0.5% | | $23.9 \pm 9.4$ | | $30.0 \pm 9.3$ | 6.1 | | $33.3 \pm 7.2$ | 9.4 |
| | | 0.35% | | $31.3 \pm 10.0$ | | $36.5 \pm 7.7$ | 5.2 | | $38.8 \pm 6.5$ | 7.5 |
| | VGG16 | 1% | 67.8 | $7.0 \pm 3.5$ | 53.6 | $11.2 \pm 4.4$ | 4.2 | 55.2 | $13.6 \pm 5.2$ | 6.6 |
| | | 0.5% | | $22.4 \pm 7.0$ | | $32.4 \pm 7.3$ | 10.0 | | $35.9 \pm 6.2$ | 13.5 |
| | | 0.35% | | $31.1 \pm 7.2$ | | $39.4 \pm 6.3$ | 8.3 | | $42.4 \pm 4.9$ | 11.3 |
| | VGG19 | 0.5% | 67.8 | $34.0 \pm 9.6$ | 59.4 | $50.2 \pm 3.1$ | 16.2 | 58.7 | $49.1 \pm 3.5$ | 15.1 |
| | | 0.35% | | $42.1 \pm 9.4$ | | $53.1 \pm 2.3$ | 11.0 | | $52.0 \pm 3.1$ | 9.9 |

[Note] S.M.: *source model, used for training generators*, T.M.: *target model, used for testing generators*, B.E.R.: *the bit error rate of the target model*, C.A. (%): *clean accuracy*, P.A. (%): *perturbed accuracy*, NF: *NeuralFuse*, and R.P.: *total recover percentage of P.A. (NF) v.s. P.A.*

Table 19: Transfer results on CIFAR-100: NeuralFuse trained on S.M. with 0.5% B.E.R.

| S.M. | T.M. | B.E.R. | C.A. | P.A. | ConvL (0.5%) | | | UNetL (0.5%) | | |
|---|---|---|---|---|---|---|---|---|---|---|
| | | | | | C.A. (NF) | P.A. (NF) | R.P. | C.A. (NF) | P.A. (NF) | R.P. |
| ResNet18 | ResNet18 | 0.35% | 73.7 | $31.4 \pm 7.6$ | 65.2 | $47.7 \pm 4.9$ | 16.3 | 66.2 | $49.2 \pm 4.1$ | 17.8 |
| | ResNet50 | 0.5% | 73.5 | $21.3 \pm 7.0$ | 62.5 | $24.0 \pm 9.9$ | 2.8 | 63.5 | $26.4 \pm 9.1$ | 5.1 |
| | | 0.35% | | $35.7 \pm 8.6$ | | $36.3 \pm 8.9$ | 0.6 | | $39.4 \pm 8.1$ | 3.7 |
| | VGG11 | 0.5% | 64.8 | $23.9 \pm 9.4$ | 59.2 | $33.0 \pm 9.8$ | 9.2 | 61.1 | $34.2 \pm 9.8$ | 10.3 |
| | | 0.35% | | $31.3 \pm 10.0$ | | $40.4 \pm 8.7$ | 9.1 | | $41.4 \pm 9.0$ | 10.1 |
| | VGG16 | 0.5% | 67.8 | $22.4 \pm 7.0$ | 59.5 | $34.7 \pm 8.0$ | 12.3 | 61.4 | $37.5 \pm 6.8$ | 15.2 |
| | | 0.35% | | $31.1 \pm 7.2$ | | $42.9 \pm 6.0$ | 11.8 | | $45.3 \pm 4.9$ | 14.2 |
| | VGG19 | 0.5% | 67.8 | $34.0 \pm 9.6$ | 61.6 | $43.7 \pm 6.2$ | 9.6 | 62.0 | $45.0 \pm 6.3$ | 11.0 |
| | | 0.35% | | $42.1 \pm 9.4$ | | $49.0 \pm 5.5$ | 6.8 | | $50.5 \pm 5.3$ | 8.3 |
| VGG19 | ResNet18 | 0.5% | 73.7 | $20.9 \pm 7.4$ | 66.1 | $24.9 \pm 6.7$ | 4.0 | 67.8 | $27.7 \pm 6.8$ | 6.8 |
| | | 0.35% | | $31.4 \pm 7.6$ | | $34.4 \pm 5.4$ | 3.0 | | $38.1 \pm 5.6$ | 6.7 |
| | ResNet50 | 0.5% | 73.5 | $21.3 \pm 7.0$ | 66.2 | $22.7 \pm 7.8$ | 1.4 | 66.7 | $25.4 \pm 8.0$ | 4.2 |
| | | 0.35% | | $35.7 \pm 8.6$ | | $35.5 \pm 7.7$ | -0.2 | | $38.8 \pm 7.5$ | 3.2 |
| | VGG11 | 0.5% | 64.8 | $23.9 \pm 9.4$ | 59.9 | $29.3 \pm 10$ | 5.4 | 61.0 | $31.2 \pm 9.8$ | 7.4 |
| | | 0.35% | | $31.3 \pm 10.0$ | | $36.6 \pm 9.5$ | 5.3 | | $38.1 \pm 9.0$ | 6.8 |
| | VGG16 | 0.5% | 67.8 | $22.4 \pm 7.0$ | 62.5 | $30.8 \pm 7.3$ | 8.4 | 62.6 | $33.0 \pm 7.3$ | 10.7 |
| | | 0.35% | | $31.1 \pm 7.2$ | | $40.0 \pm 6.5$ | 8.9 | | $42.5 \pm 5.9$ | 11.3 |
| | VGG19 | 0.35% | 67.8 | $42.1 \pm 9.4$ | 65.6 | $52.0 \pm 6.2$ | 9.8 | 65.5 | $52.6 \pm 6.1$ | 10.4 |

[Note] S.M.: *source model, used for training generators*, T.M.: *target model, used for testing generators*, B.E.R.: *the bit error rate of the target model*, C.A. (%): *clean accuracy*, P.A. (%): *perturbed accuracy*, NF: *NeuralFuse*, and R.P.: *total recover percentage of P.A. (NF) v.s. P.A.*

## H.4   GENERATOR ENSEMBLING

To improve the transferability performance on cross-architecture cases (e.g., using ResNet-based models as surrogate models to train NeuralFuse and then transfer NeuralFuse to VGG-based target models), we try to adopt ensemble surrogate models to train our NeuralFuse. The experimental results are shown in Table 20. We use the same experimental settings mentioned in Table 1 but change one source model (e.g., ResNet18 or VGG19) into two (ResNet18 with VGG19) for training. The results show that the overall performance is better than the results shown in Table 1, which means ensemble-based training can easily solve the performance degradation on cross-architecture target models.

Table 20: Transfer results on CIFAR-10: NeuralFuse trained on two S.M. with 1.5% B.E.R.

| S.M. | T.M. | B.E.R. | C.A. | P.A. | ConvL (1.5%) | | | UNetL (1.5%) | | |
| | | | | | C.A. (NF) | P.A. (NF) | R.P. | C.A. (NF) | P.A. (NF) | R.P. |
|---|---|---|---|---|---|---|---|---|---|---|
| ResNet18 + VGG19 | ResNet18 | 1% | 92.6 | $38.9 \pm 12.4$ | 89.4 | $88.1 \pm 1.0$ | 49.2 | 86.3 | $85.4 \pm 0.5$ | 46.5 |
| | | 0.5% | | $70.1 \pm 11.6$ | | $89.2 \pm 0.2$ | 19.1 | | $86.1 \pm 0.2$ | 16.0 |
| | ResNet50 | 1% | 92.6 | $26.1 \pm 9.4$ | 89.3 | $44.0 \pm 22$ | 17.9 | 86.1 | $50.9 \pm 20$ | 24.8 |
| | | 0.5% | | $61.0 \pm 10.3$ | | $80.3 \pm 6.7$ | 19.3 | | $78.6 \pm 3.9$ | 17.6 |
| | VGG11 | 1% | 88.4 | $42.2 \pm 11.6$ | 89.1 | $77.0 \pm 5.6$ | 34.8 | 85.9 | $82.3 \pm 4.1$ | 40.1 |
| | | 0.5% | | $63.6 \pm 9.3$ | | $87.5 \pm 1.6$ | 23.9 | | $85.0 \pm 0.6$ | 21.4 |
| | VGG16 | 1% | 90.3 | $35.7 \pm 7.9$ | 89.1 | $80.5 \pm 8.6$ | 44.8 | 85.7 | $81.4 \pm 5.5$ | 45.7 |
| | | 0.5% | | $66.6 \pm 8.1$ | | $88.2 \pm 0.7$ | 21.6 | | $85.0 \pm 0.7$ | 18.4 |
| | VGG19 | 1% | 90.5 | $36.0 \pm 12.0$ | 89.2 | $75.1 \pm 17$ | 39.1 | 86.1 | $83.0 \pm 3.4$ | 47.0 |
| | | 0.5% | | $64.2 \pm 12.4$ | | $89.0 \pm 0.2$ | 24.8 | | $85.9 \pm 0.4$ | 21.7 |

[Note] S.M.: *source model, used for training generators*, T.M.: *target model, used for testing generators*, B.E.R.: *the bit error rate of the target model*, C.A. (%): *clean accuracy*, P.A. (%): *perturbed accuracy*, NF: *NeuralFuse*, and R.P.: *total recover percentage of P.A. (NF) v.s. P.A.*

## I   NEURALFUSE ON REDUCED-PRECISION QUANTIZATION AND RANDOM BIT ERRORS

As mentioned in Sec. 4.6, we explore the robustness of NeuralFuse to low-precision quantization of model weights and consider the case of random bit errors. Here, we demonstrate that Neural-Fuse can recover not only the accuracy drop due to reduced precision, but also the drop caused by low-voltage-induced bit errors (0.5% B.E.R.) under low precision. We selected two NeuralFuse generators (ConvL and UNetL) for our experiments, and these generators were trained with the corresponding base models (ResNet18 and VGG19) at 1% B.E.R. (CIFAR-10, GTSRB) and 0.5% B.E.R. (ImageNet-10). The experimental results are shown as follows: CIFAR-10 (Sec. I.1), GT-SRB (Sec. I.2), and ImageNet-10 (Sec. I.3). Similarly, for ease of comparison, we visualize the experimental results in the figures below each table. Our results show that NeuralFuse can consistently perform well in low-precision regimes as well as recover the low-voltage-induced accuracy drop.

### I.1   CIFAR-10

Table 21: Reduced-precision Quantization and with 0.5% B.E.R. on CIFAR-10 pre-trained models.

| Base Model | #Bits | C.A. | P.A. | ConvL (1%) | | | UNetL (1%) | | |
| | | | | C.A. (NF) | P.A. (NF) | R.P. | C.A. (NF) | P.A. (NF) | R.P. |
|---|---|---|---|---|---|---|---|---|---|
| ResNet18 | 8 | 92.6 | $70.1 \pm 11.6$ | 89.8 | $89.5 \pm 0.2$ | 19.4 | 86.6 | $86.2 \pm 0.3$ | 16.1 |
| | 7 | 92.5 | $68.8 \pm 10.4$ | 89.8 | $89.5 \pm 1.7$ | 20.7 | 86.5 | $86.0 \pm 0.5$ | 17.2 |
| | 6 | 92.6 | $68.4 \pm 11.2$ | 89.7 | $89.5 \pm 0.2$ | 21.1 | 86.6 | $85.9 \pm 0.3$ | 17.5 |
| | 5 | 92.4 | $52.7 \pm 14.1$ | 89.7 | $90.0 \pm 0.7$ | 37.3 | 86.5 | $85.5 \pm 0.8$ | 32.8 |
| | 4 | 91.8 | $26.3 \pm 12.7$ | 89.8 | $58.7 \pm 24.5$ | 32.4 | 86.6 | $64.9 \pm 22.5$ | 38.6 |
| | 3 | 84.8 | $11.3 \pm 1.8$ | 89.8 | $12.8 \pm 5.8$ | 1.5 | 86.0 | $14.8 \pm 10.0$ | 3.5 |
| | 2 | 10.0 | $10.0 \pm 0.0$ | 10.0 | $10.0 \pm 0.0$ | 0.0 | 10.0 | $10.0 \pm 0.0$ | 0.0 |
| VGG19 | 8 | 90.5 | $64.2 \pm 12.4$ | 89.8 | $89.6 \pm 8.7$ | 25.4 | 87.4 | $86.8 \pm 0.4$ | 22.6 |
| | 7 | 90.3 | $66.5 \pm 8.5$ | 89.8 | $89.6 \pm 0.2$ | 23.1 | 87.4 | $86.7 \pm 0.3$ | 20.2 |
| | 6 | 90.1 | $59.8 \pm 13.2$ | 89.9 | $89.4 \pm 3.8$ | 29.6 | 87.4 | $86.4 \pm 0.7$ | 26.6 |
| | 5 | 90.2 | $37.7 \pm 14.1$ | 89.8 | $78.0 \pm 15.8$ | 40.3 | 87.2 | $79.8 \pm 0.8$ | 42.1 |
| | 4 | 87.5 | $14.7 \pm 6.0$ | 89.8 | $27.8 \pm 18.9$ | 13.1 | 87.2 | $34.4 \pm 20.5$ | 19.7 |
| | 3 | 78.3 | $10.5 \pm 1.5$ | 89.7 | $10.9 \pm 2.6$ | 0.4 | 86.8 | $11.0 \pm 2.9$ | 0.5 |
| | 2 | 10.0 | $10.0 \pm 0.0$ | 10.0 | $10.0 \pm 0.0$ | 0.0 | 10.0 | $10.0 \pm 0.0$ | 0.0 |

[Note] C.A. (%): *clean accuracy*, P.A. (%): *perturbed accuracy*, NF: *NeuralFuse*, and R.P.: *total recover percentage of P.A. (NF) v.s. P.A.*

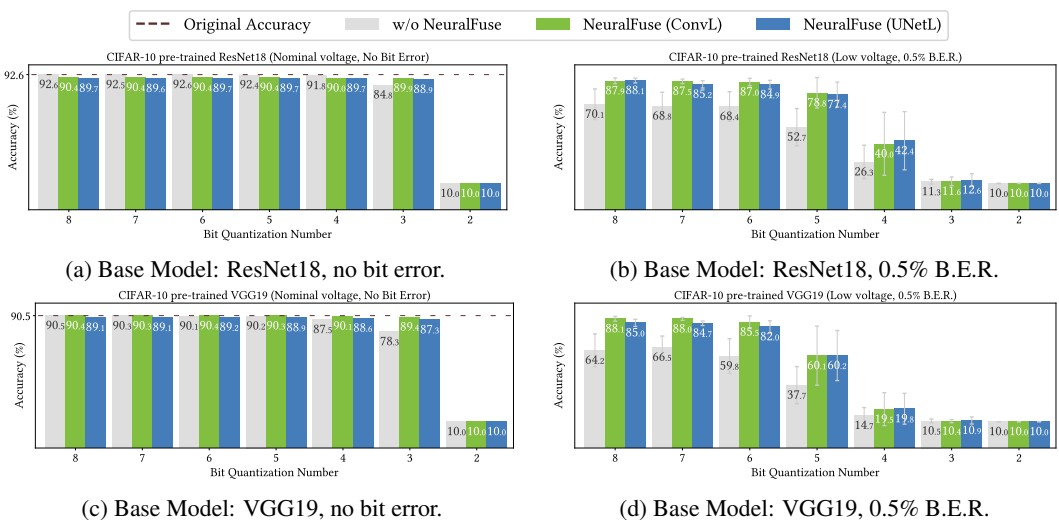

(a) Base Model: ResNet18, no bit error.

(b) Base Model: ResNet18, 0.5% B.E.R.

(c) Base Model: VGG19, no bit error.

(d) Base Model: VGG19, 0.5% B.E.R.

Figure 12: Results of Reduced-precision and bit errors (0.5%) on CIFAR-10 pre-trained base models.

## I.2 GTSRB

Table 22: Reduced-precision Quantization and with 0.5% B.E.R. on GTSRB pre-trained models.

| Base Model | #Bits | C.A. | P.A. | ConvL (1%) | | | UNetL (1%) | | |
|---|---|---|---|---|---|---|---|---|---|
| | | | | C.A. (NF) | P.A. (NF) | R.P. | C.A. (NF) | P.A. (NF) | R.P. |
| ResNet18 | 8 | 95.5 | $75.2 \pm 12.7$ | 95.7 | $95.3 \pm 0.5$ | 20.1 | 96.2 | $95.7 \pm 0.3$ | 20.5 |
| | 7 | 95.5 | $69.5 \pm 10.6$ | 95.7 | $95.3 \pm 0.3$ | 25.8 | 96.2 | $95.9 \pm 0.3$ | 26.4 |
| | 6 | 95.4 | $67.2 \pm 14.4$ | 95.7 | $95.2 \pm 0.5$ | 28.0 | 96.2 | $95.7 \pm 0.5$ | 28.5 |
| | 5 | 95.4 | $48.6 \pm 18.2$ | 95.8 | $92.6 \pm 5.1$ | 44.0 | 96.2 | $94.8 \pm 2.5$ | 46.2 |
| | 4 | 92.6 | $24.6 \pm 9.8$ | 95.9 | $75.6 \pm 16.2$ | 51.0 | 96.2 | $86.6 \pm 9.5$ | 62.0 |
| | 3 | 67.7 | $5.3 \pm 3.5$ | 95.4 | $18.4 \pm 15.3$ | 13.1 | 96.2 | $25.3 \pm 22.5$ | 20.0 |
| | 2 | 3.8 | $3.8 \pm 0.0$ | 4.1 | $3.8 \pm 0.0$ | 0.0 | 3.8 | $3.8 \pm 0.0$ | 0.0 |
| VGG19 | 8 | 95.5 | $69.1 \pm 11.1$ | 96.0 | $94.0 \pm 2.2$ | 24.9 | 95.4 | $93.9 \pm 2.1$ | 24.8 |
| | 7 | 95.6 | $66.1 \pm 14.8$ | 96.0 | $92.2 \pm 5.7$ | 26.1 | 95.4 | $92.6 \pm 3.7$ | 26.5 |
| | 6 | 95.3 | $64.2 \pm 8.4$ | 96.0 | $92.2 \pm 5.7$ | 28.0 | 95.4 | $92.3 \pm 2.3$ | 28.1 |
| | 5 | 95.2 | $48.2 \pm 14.0$ | 96.0 | $92.2 \pm 5.7$ | 44.0 | 95.4 | $86.2 \pm 8.4$ | 38.0 |
| | 4 | 92.0 | $18.2 \pm 14.3$ | 93.0 | $92.2 \pm 5.7$ | 74.0 | 95.0 | $49.6 \pm 22.8$ | 31.4 |
| | 3 | 60.0 | $2.0 \pm 0.9$ | 87.3 | $92.2 \pm 5.7$ | 90.2 | 87.2 | $1.7 \pm 0.9$ | -0.3 |
| | 2 | 5.9 | $3.8 \pm 0.0$ | 5.9 | $3.8 \pm 0.0$ | 0.0 | 5.9 | $3.8 \pm 0.0$ | 0.0 |

[Note] C.A. (%): *clean accuracy*, P.A. (%): *perturbed accuracy*, NF: *NeuralFuse*, and R.P.: *total recover percentage of P.A. (NF) v.s. P.A.*

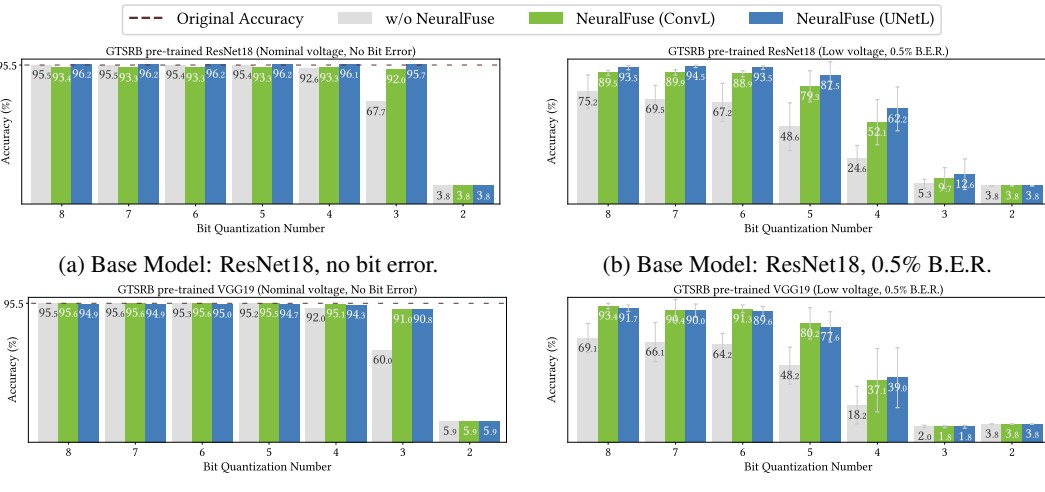

(a) Base Model: ResNet18, no bit error.

(b) Base Model: ResNet18, 0.5% B.E.R.

(c) Base Model: VGG19, no bit error.

(d) Base Model: VGG19, 0.5% B.E.R.

Figure 13: Results of Reduced-precision and bit errors (0.5%) on GTSRB pre-trained base models.

## I.3 IMAGENET-10

Table 23: Reduced-precision Quantization and with 0.5% B.E.R. on ImageNet-10 pre-trained models.

| Base Model | #Bits | C.A. | P.A. | ConvL (0.5%) | | | UNetL (0.5%) | | |
|---|---|---|---|---|---|---|---|---|---|
| | | | | C.A. (NF) | P.A. (NF) | R.P. | C.A. (NF) | P.A. (NF) | R.P. |
| ResNet18 | 8 | 92.2 | 72.3 ± 7.0 | 94.0 | 88.0 ± 2.0 | 15.7 | 94.0 | 88.1 ± 1.4 | 15.8 |
| | 7 | 92.4 | 70.6 ± 13.0 | 94.2 | 86.7 ± 4.1 | 16.1 | 93.6 | 87.8 ± 3.5 | 17.2 |
| | 6 | 92.4 | 68.9 ± 9.9 | 94.2 | 85.1 ± 4.8 | 16.2 | 93.6 | 86.4 ± 3.7 | 17.5 |
| | 5 | 91.0 | 60.9 ± 13.0 | 94.2 | 82.5 ± 6.8 | 21.6 | 94.0 | 83.2 ± 5.9 | 22.3 |
| | 4 | 91.4 | 47.4 ± 9.8 | 93.8 | 68.6 ± 9.8 | 21.2 | 92.6 | 68.7 ± 9.2 | 21.3 |
| | 3 | 85.2 | 28.8 ± 11.8 | 89.2 | 44.1 ± 14.0 | 15.3 | 89.4 | 42.7 ± 14.2 | 13.9 |
| | 2 | 10.0 | 10.0 ± 0.0 | 10.0 | 10.0 ± 0.0 | 0.0 | 10.0 | 10.0 ± 0.0 | 0.0 |
| VGG19 | 8 | 92.4 | 37.2 ± 11.0 | 91.4 | 75.5 ± 8.8 | 38.3 | 89.4 | 77.9 ± 6.1 | 40.7 |
| | 7 | 92.0 | 27.3 ± 6.6 | 91.2 | 59.3 ± 13.0 | 32.0 | 89.4 | 65.4 ± 10.0 | 38.1 |
| | 6 | 92.4 | 27.9 ± 6.4 | 91.0 | 59.7 ± 11.8 | 31.8 | 89.4 | 64.9 ± 9.9 | 37.0 |
| | 5 | 92.0 | 15.1 ± 4.4 | 91.6 | 23.1 ± 0.7 | 8.0 | 89.0 | 27.9 ± 8.8 | 12.8 |
| | 4 | 89.4 | 12.2 ± 2.7 | 90.8 | 14.0 ± 4.3 | 1.8 | 89.6 | 14.6 ± 4.9 | 2.4 |
| | 3 | 46.8 | 9.9 ± 0.5 | 83.2 | 10.4 ± 0.6 | 0.5 | 84.2 | 9.9 ± 0.7 | 0.0 |
| | 2 | 10.0 | 10.0 ± 0.0 | 10.0 | 10.0 ± 0.0 | 0.0 | 10.0 | 10.0 ± 0.0 | 0.0 |

[Note] C.A. (%): *clean accuracy*, P.A. (%): *perturbed accuracy*, NF: *NeuralFuse*, and R.P.: *total recover percentage of P.A. (NF) v.s. P.A.*

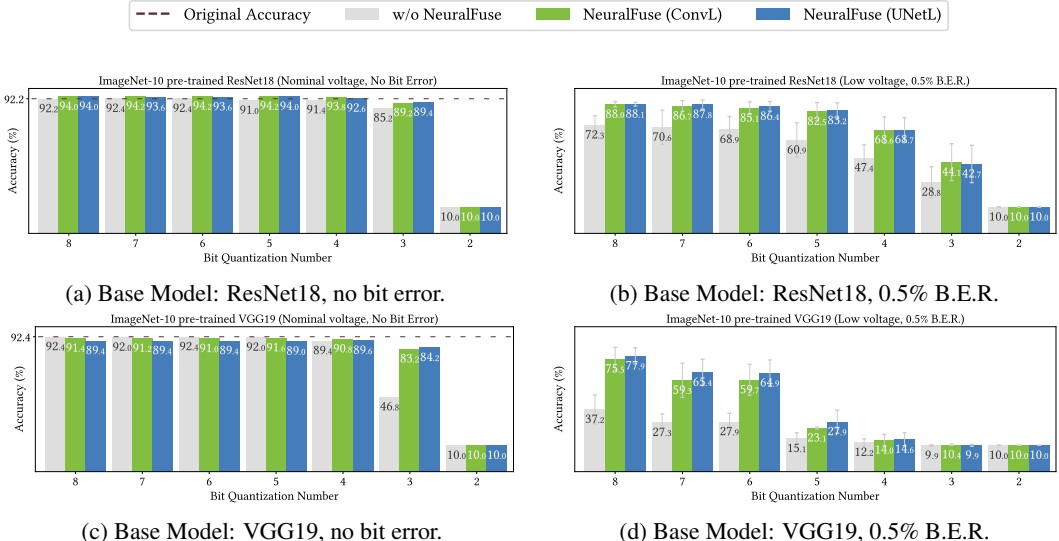

(a) Base Model: ResNet18, no bit error.

(b) Base Model: ResNet18, 0.5% B.E.R.

(c) Base Model: VGG19, no bit error.

(d) Base Model: VGG19, 0.5% B.E.R.

Figure 14: Results of Reduced-precision and bit errors (0.5%) on ImageNet-10 pre-trained base models.

## J    DATA EMBEDDINGS VISUALIZATION

To further understand how our proposed NeuralFuse works, we visualize the output distribution from the final linear layer of the base models and project the results onto the 2D space using t-SNE (van der Maaten & Hinton, 2008). Figure 15 shows the output distribution from ResNet18 (trained on CIFAR-10) under a 1% bit error rate. We chose two generators that have similar architecture: ConvL and ConvS, for this experiment. We can observe that: (a) The output distribution of the clean model without NeuralFuse can be grouped into 10 classes denoted by different colors. (b) The output distribution of the perturbed model under a 1% bit error rate without NeuralFuse shows mixed representations and therefore degraded accuracy. (c) The output distribution of the clean model with ConvL shows that applying NeuralFuse will not hurt the prediction of the clean model too much (i.e., it retains high accuracy in the regular voltage setting). (d) The output distribution of the perturbed model with ConvL shows high separability (and therefore high perturbed accuracy) as opposed to (b). (e)/(f) shows the output distribution of the clean/perturbed model with ConvS. For both (e) and (f), we can see nosier clustering when compared to (c) and (d), which means the degraded performance of ConvS compared to ConvL. The visualization validates that NeuralFuse can help retain good data representations under random bit errors and that larger generators in NeuralFuse have better performance than smaller ones.

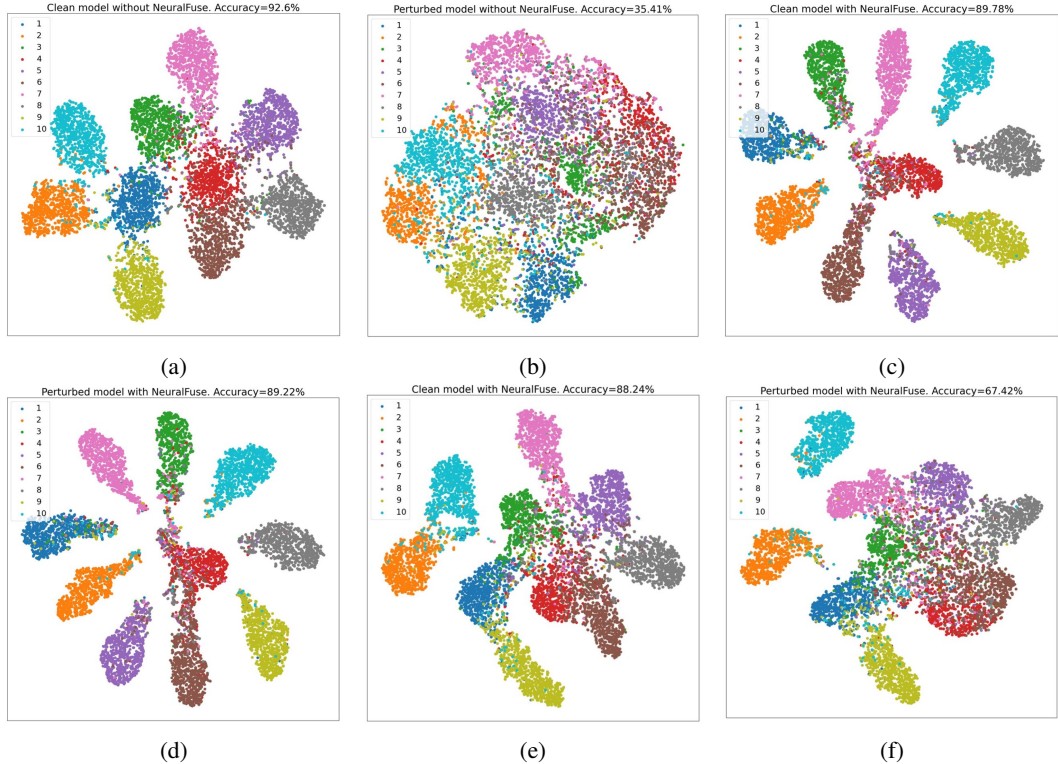

Figure 15: t-SNE results for ResNet18 trained by CIFAR-10 under 1% of bit error rate. (a) Clean model. (b) Perturbed model. (c) Clean model with ConvL. (d) Perturbed model with ConvL. (e) Clean model with ConvS. (f) Perturbed model with ConvS.

## K    QUALITATIVE ANALYSIS OF TRANSFORMED INPUTS

In this section, we conduct a qualitative study to visualize the images which are transformed by NeuralFuse, and then present some properties of these images. We adopt six different architectures of NeuralFuse generators trained with ResNet18 under a 1% bit error rate. In Figure 16(a), we show several images from the truck class in CIFAR-10. We observe that different images in the same class transformed by the same NeuralFuse will exhibit a similar pattern. For example, the

patterns contain several circles, which may symbolize the wheels of the trucks. In Figure 16(b), we show several images of a traffic sign category (No Overtaking) in GTSRB. We also oversee that the transformed images contain similar patterns. In particular, in GTSRB, NeuralFuse will generate patterns that highlight the shape of the sign with a green background, even if the original images are of a dark background and under different lighting conditions.

In Figure 17, we show the images from ten different classes in CIFAR-10 and GTSRB separately. The transformed images have distinct patterns for each class. Therefore, we speculate that Neural-Fuse effectively transforms images to some class-specific patterns such that the associated features are robust to random bit errors and can be easily recognizable by the base model in low-voltage settings.

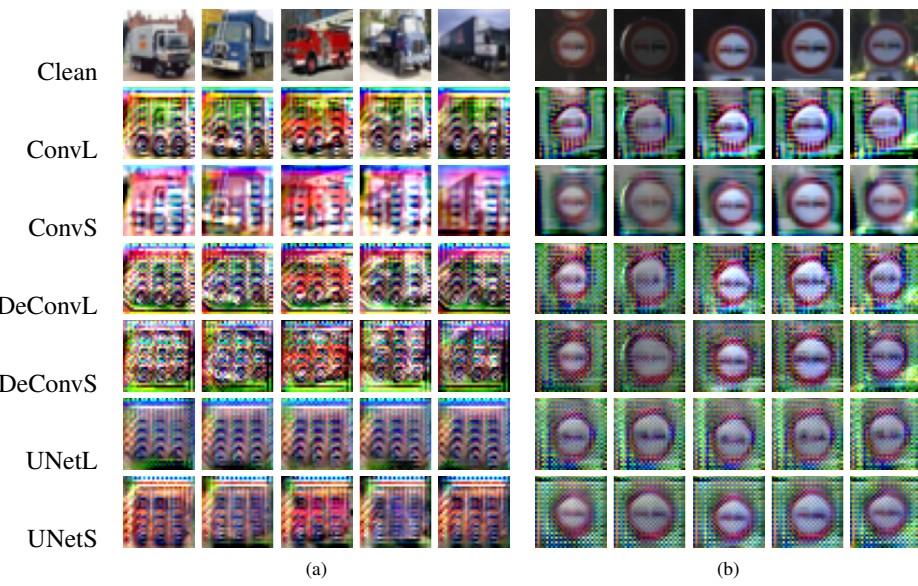

Figure 16: Visualization results of the transformed images from six different NeuralFuse generators trained with ResNet18 under $1\%$ bit error rate. (a) *Truck* class in CIFAR-10. (b) *No Overtaking (general)* sign in GTSRB.

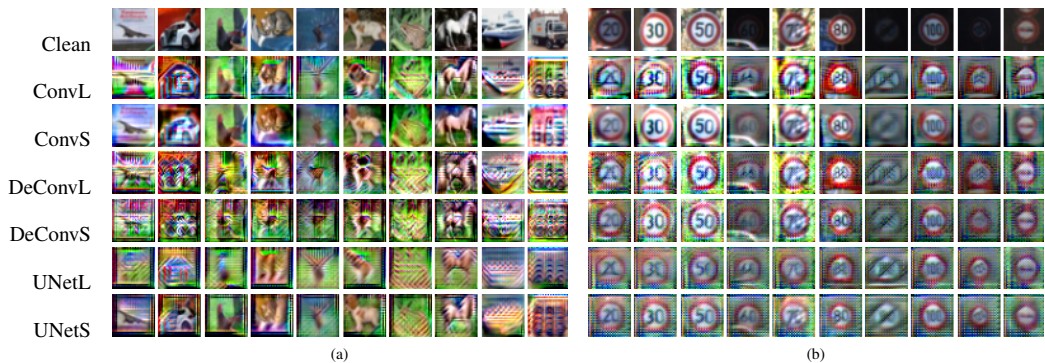

Figure 17: Visualization results of the transformed images from six different NeuralFuse generators trained by ResNet18 with $1\%$ bit error rate. (a) Ten different classes sampled from CIFAR-10. (b) Ten different traffic signs sampled from GTSRB.

In Figure 18, we show several images from the apple class in CIFAR-100. We observe that the different images transformed by the same NeuralFuse will provide the similar patterns. This observation is similar to CIFAR-10 and GTSRB mentioned above. In Figure 19, we show more different classes and their corresponding transformed results.

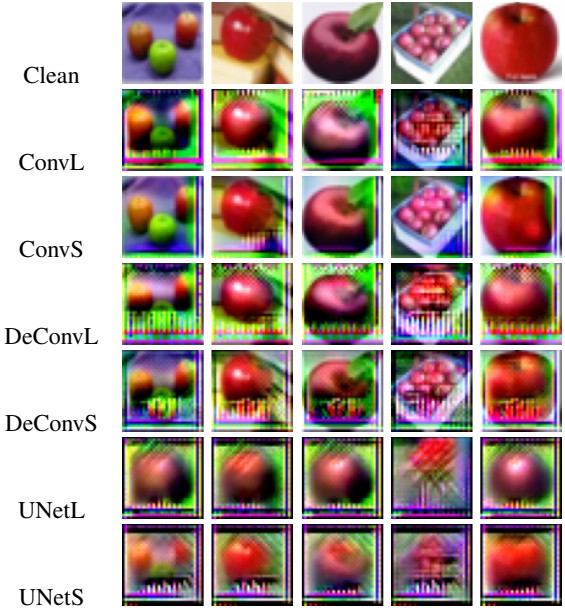

Figure 18: Visualization results of the transformed images on CIFAR-100 from six different NeuralFuse generators trained with ResNet18 under $1\%$ of bit error rate.

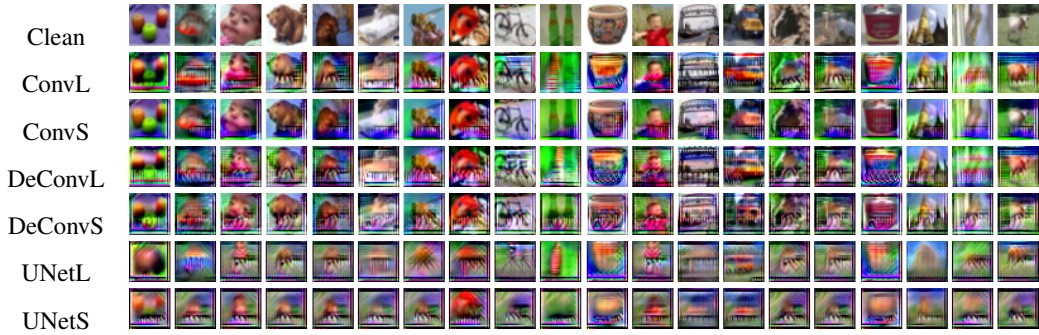

Figure 19: Visualization results of twenty different classes of the transformed images from CIFAR-100 made by six different NeuralFuse generators, which are trained with ResNet18 under $1\%$ of bit error rate.

## L  ADDITIONAL EXPERIMENTS ON ADVERSARIAL TRAINING

Adversarial training is a common strategy to derive a robust neural network against certain perturbations. By training the generator using adversarial training proposed in Stutz et al. (2021), we report its performance against low voltage-induced bit errors. We use ConvL as the generator and ResNet18 as the base model, trained on CIFAR-10. Furthermore, we explore different $K$ flip bits as the perturbation on weights of the base model during adversarial training, and then for evaluation, the trained-generator will be applied against 1% of bit errors rate on the base model. The results are shown in Table 24. After careful tuning of hyperparameters, we find that we are not able to obtain satisfactory recovery when adopting adversarial training. Empirically, we argue that adversarial training may not be suitable for training generator-based methods.

Table 24: Performance of the generator trained by adversarial training under K flip bits on ResNet18 with CIFAR-10. The results show that the generator trained by adversarial training cannot achieve high accuracy against bit errors under 1% bit error rate.

| K-bits | C.A. | P.A. | C.A. (NF) | P.A. (NF) | R.P. |
|---|---|---|---|---|---|
| 100 | | | 92.4 | $38.3 \pm 12.1$ | -0.6 |
| 500 | | | 92.1 | $38.7 \pm 12.5$ | -0.2 |
| 5,000 | 92.6 | $38.9 \pm 12.4$ | 92.6 | $38.9 \pm 12.5$ | 0 |
| 20,000 | | | 60.1 | $23.0 \pm 8.1$ | -16 |
| 100,000 | | | 71.1 | $23.6 \pm 6.6$ | -16 |

[Note] C.A. (%): *clean accuracy*, P.A. (%): *perturbed accuracy*, NF: *NeuralFuse*, and R.P.: *total recover percentage of P.A. (NF) v.s. P.A.*

## M  ADDITIONAL EXPERIMENTS ON ROBUST MODEL TRAINED WITH ADVERSARIAL WEIGHT PERTURBATION WITH NEURALFUSE

Previously, Wu et al. proposed that one could obtain a more robust model via adversarial weight perturbation (Wu et al., 2020). To seek whether such models could also be robust to random bit errors, we conducted an experiment on CIFAR-10 with the proposed adversarially trained PreAct ResNet18. The experimental results are shown in Table 25. We find that the average perturbed accuracy is 23% and 63.2% for PreAct ResNet18 under 1% and 0.5% B.E.R., respectively. This result is lower than 38.9% and 70.1% from ResNet18 in Table 11, indicating their poor generalization ability against random bit errors. Nevertheless, when equipped NeuralFuse on the perturbed model, we could still witness a significant recover percentage under both 1% and 0.5% B.E.R. This result further demonstrates that NeuralFuse could be adapted to various models (i.e., trained in different learning algorithms).

Table 25: Performance of NeuralFuse trained with rubust CIFAR-10 pre-trained PreAct ResNet18. The results show that NeuralFuse can be used together with a robust model and further improve perturbed accuracy under both 1% and 0.5% B.E.R.

| Base Model | B.E.R. | NF | C.A. | P.A. | C.A. (NF) | P.A. (NF) | R.P. |
|---|---|---|---|---|---|---|---|
| PreAct ResNet18 | 1% | ConvL | 89.7 | $23.0 \pm 9.3$ | 87.6 | $53.7 \pm 26$ | 30.7 |
| | | ConvS | | | 83.1 | $34.6 \pm 15$ | 11.6 |
| | | DeConvL | | | 87.7 | $55.4 \pm 27$ | 32.4 |
| | | DeConvS | | | 82.9 | $32.4 \pm 14$ | 9.4 |
| | | UNetL | | | 86.1 | $60.4 \pm 28$ | 37.4 |
| | | UNetS | | | 80.4 | $51.9 \pm 24$ | 28.9 |
| | 0.5% | ConvL | 89.7 | $63.2 \pm 8.7$ | 89.2 | $87.8 \pm 1.1$ | 24.6 |
| | | ConvS | | | 89.2 | $74.0 \pm 6.5$ | 10.8 |
| | | DeConvL | | | 89.0 | $87.4 \pm 1.1$ | 24.2 |
| | | DeConvS | | | 89.9 | $74.4 \pm 7.0$ | 11.2 |
| | | UNetL | | | 87.5 | $85.9 \pm 0.8$ | 22.7 |
| | | UNetS | | | 88.2 | $80.4 \pm 3.9$ | 17.2 |

[Note] B.E.R.: *the bit error rate of the base model*, NF: *NeuralFuse*, C.A. (%): *clean accuracy*, P.A. (%): *perturbed accuracy*, and R.P.: *total recover percentage of P.A. (NF) v.s. P.A.*

## N  INFERENCE LATENCY OF NEURALFUSE

In Table 26, we report the latency (batch_size=1, CIFAR-10/ImageNet-10 testing dataset) of utilizing the different NeuralFuse generators with two different base models, ResNet18 and VGG19. We can see that although NeuralFuse indeed brings a certain degree of extra latency, we argue that this is an unavoidable factor; however, since the latency is measured on a general-purpose GPU (i.e. V100), when the base model and NeuralFuse are deployed on a custom accelerator, we believe this delay will be further reduced.

Table 26: The Inference Latency of base model and base model with NeuralFuse.

|  | ResNet18 (CIFAR-10) | VGG19 (CIFAR-10) | ResNet18 (ImageNet-10) | VGG19 (ImageNet-10) |
|---|---|---|---|---|
| Base Model | 5.84 ms | 5.32 ms | 6.21 ms | 14.34 ms |
| + ConvL | 9.37 ms (+3.53) | 8.96 ms (+3.64) | 10.51 ms (+4.3) | 17.66 ms (+3.32) |
| + ConvS | 7.86 ms (+2.02) | 7.40 ms (+2.08) | 8.28 ms (+2.07) | 16.72 ms (+2.38) |
| + DeConvL | 9.18 ms (+3.34) | 8.59 ms (+3.27) | 10.07 ms (+3.86) | 17.24 ms (+2.90) |
| + DeConvS | 7.49 ms (+1.65) | 7.04 ms (+1.72) | 7.79 ms (+1.58) | 15.67 ms (+1.33) |
| + UNetL | 10.69 ms (+4.85) | 10.06 ms (+4.74) | 11.14 ms (+4.93) | 18.54 ms (+4.20) |
| + UNetS | 10.63 ms (+4.79) | 10.13 ms (+4.81) | 11.36 ms (+5.15) | 18.60 ms (+4.26) |

## O  DISCUSSION FOR REAL-WORLD APPLICATION OR POTENTIAL USE CASES

In this section, we provide some possible real-world applications for using NeuralFuse under low-voltage regimes. Previous works have pointed out some possible scenarios that suffer from energy concerns and hence need some strategies to reduce energy consumption. For example, in Yang et al. (2017; 2019a), the authors mention that due to the high computation cost of CNN processing and some DNN-based vision algorithms, they will incur high energy consumption. This will significantly reduce the battery life of battery-powered devices, indirectly impacting the user experience of the devices. Therefore, to avoid the aforementioned issues, we can mitigate the device's energy consumption by lowering the operating voltage and then incorporating NeuralFuse to recover model performance, reducing the side effects caused by low voltage.

