# OpenReview forum: "NeuralFuse: Learning to Recover the Accuracy of Access-Limited Neural Network Inference in Low-Voltage Regimes"
_ICLR.cc/2024/Conference — Submitted to ICLR 2024_

### Official Review · Reviewer_7gLE · 2023-10-29

**Soundness:** 3 good
**Presentation:** 3 good
**Contribution:** 3 good
**Rating:** 5
**Confidence:** 4

**Summary:**

This paper proposes NeuralFuse, a model-agnostic approach that learns input transformations to generate error-resistant data representations. NeuralFuse dynamically adds a correction term to the model input to protect the DNNs in both nominal and low-voltage scenarios and can be applied to DNNs with limited access. Experimental results show that NeuralFuse can reduce SRAM memory access energy by up to 20-30% while recovering accuracy by up to 57% at a 1% bit error rate.

**Strengths:**

Strength:

1.	The idea is quite interesting. Without error-aware training (adversarial training), it learns input-dependent, model-agnostic calibrator for the model input, the DNN’s accuracy can be protected.

2.	The proposed neural network can protect the DNN accuracy while still showing energy efficiency benefits.

3.	It thoroughly investigates the transferability to different error rates, model architecture, and quantization bitwidth.

**Weaknesses:**

Weakness:
1.	The transferability on different error rate and model size is not very good, according to Table 1, which means re-training is still required for different model/dataset/SRAM voltages.

2.	The energy saving is only ~20% by reducing SRAM voltage, while the accuracy drop is beyond 1%. It needs some justification on this trade-off.

3.	The method seems to be equivalent to adding extra layers in the early stage of the network and train it with noise-aware training. Why not train other layers with the memory error? It is not very intuitive that weight errors in all layers (even MSB flips) can be well protected by only changing the model input. More explanation is needed to justify this. Can we add a protector to later layers and do some calibration? Or even parallel branches? Or protect the weights loaded from memory block-wise, which can still maintain model-agnostic?

**Questions:**

listed in the weakness part.

---

> ### Author Response · Authors · 2023-11-21
> **Response to Reviewer 7gLE (1 of 2)**
>
> Dear Reviewer 7gLE,
>
> Thank you for taking the time to review our paper. We are delighted to hear that you have found our work interesting and recognized its benefit on energy efficiency. We have updated our paper based on the reviewers' comments. The following are our responses to your comments on the concerns of our paper.
>
> **[W1]** The transferability on different error rates and model sizes is not very good.
>
> **[Response]** As shown in Table 1, the performance of transferability would be better when the source model and the target model are the same or similar.  Besides, in Appendix H.4, we also report the ensemble-based results, which means that we use two surrogate models to train NeuralFuse. The experimental results show that training with ensemble surrogate models can receive better transferability than only using one mentioned in Table 1. Please note that this is also a trade-off, and one should choose similar models to transfer the NeuralFuse in a limited access scenario as much as possible.
>
> ---
> **[W2]** The energy saving is only ~20% by reducing SRAM voltage, while the accuracy drop is beyond 1%.
>
> **[Response]**  We would like to point out that the accuracy drop shown in our paper was an experimental scenario to demonstrate the efficacy of NeuralFuse. Of course, one would not want to suffer a significant drop in accuracy, but it is intrinsically a trade-off between accuracy drop and energy saving. That being said, in practice, NeuralFuse can be used to adjust the voltage level (corresponding to different BERs) to achieve the purpose of energy saving, while having acceptable accuracy, which also reflects the adaptivity and generality of NeuralFuse.
>
> ---
> **[W3.1]** The method seems to be equivalent to adding extra layers in the early stage of the network and train it with noise-aware training. Why not train other layers with the memory error?
>
> **[Response]**
> This is a great insight! In fact, in Appendix M, we already compared it to PreAct ResNet18 which is trained with adversarial weights perturbation (AWP). The authors train the network with adversarial bit flips, just like adding memory errors during training. In Table 25, the experimental results show that the AWP-trained model can not resist random bit errors (the mean perturbed accuracy is only 23% under a 1% bit error rate), which is worse than using NeuralFuse as a protector to recover the accuracy under low-voltage regimes. Besides, in Section 2, we have also discussed other error-aware training techniques such as random/adversarial bit errors training proposed by [1]. However, these methods introduce challenges in hyperparameter search and hence are difficult to train. In [2], the authors have also pointed out that this error-aware training was found ineffective for large DNNs with millions of bits. Therefore, we did not take these methods into consideration.
>
> > [1] Stutz et al. Bit error robustness for energy-efficient dnn accelerators (MLSys 2021)
> >
> > [2] Özdenizci and Robert. Improving robustness against stealthy weight bit-flip attacks by output code matching (CVPR 2022)

---

> ### Author Response · Authors · 2023-11-21
> **Response to Reviewer 7gLE (2 of 2)**
>
> **[W3.2]** It is not very intuitive that weight errors in all layers (even MSB flips) can be well protected by only changing the model input. More explanation is needed to justify this.
>
> **[Response]** In general, utilizing input transformation to improve model performance can be deemed as model reprogramming [3] or visual prompting [4], which all keep the pre-trained model frozen. Previous works have showcased its efficacy in cross-domain machine learning [4, 5, 6, 7] and improved the trustworthiness properties (e.g., fairness [8], robustness [9], privacy [10], uncertainly quantification [11], etc.) of the models. Usually, the transformation (also known as prompts in other literature) is considered to be *universal*, which means different inputs will share the same pre-trained prompts. Our work provides a first attempt to design the input-aware transformation (e.g., NeuralFuse) that should be more effective in tackling more complex problems (i.e. random bit flips).
>
> In Appendix J, we visualize the output distribution from the final linear layer of the base models and project the results onto the 2D space using t-SNE. The visualization confirms that NeuralFuse can preserve accurate data representations even in the presence of random bit errors. Additionally, it demonstrates that larger generators in NeuralFuse outperform smaller ones.
>
> ---
> **[W3.3]** Can we add a protector to later layers and do some calibration? Or even parallel branches? Or protect the weights loaded from memory block-wise, which can still maintain model-agnostic?
>
> **[Response]** We appreciate the reviewer has provided many ideas that are worth exploring. We note that we actually explored some of these ideas in our submission.
> For example, as mentioned above, in Appendix M, we have already applied NeuralFuse with AWP-Based trained Pre-Act ResNet18 (a model trained with adversarial weight perturbation techniques) to do the calibration, and we indeed can recover large amounts of accuracy that take NeuralFuse as protector. Given that our work aims to open new paradigms for addressing low-voltage-induced bit errors (mogel-agnostic and can be deployed in a plug-and-play manner), the suggested ideas, while worthy of studying, are considered to be beyond the scope of our research. We will open-source our code to facilitate future studies and research exploration.
>
> >[3] Elsayed et al. Adversarial Reprogramming of Neural Networks (ICLR 2019)
> >
> >[4] Bahng et al. Exploring Visual Prompts for Adapting Large-Scale Models
> >
> >[5] Chen. Model Reprogramming: Resource-Efficient Cross-Domain Machine Learning
> >
> >[6] Bar et al. Visual Prompting via Image Inpainting (NeurIPS 2022)
> >
> >[7] Khattak et al. MaPLe: Multi-modal Prompt Learning (CVPR 2023)
> >
> >[8] Zhang et al. Fairness Reprogramming (NeurIPS 2022)
> >
> >[9] Chen et al. Visual Prompting for Adversarial Robustness (ICASSP 2023)
> >
> >[10] Li et al. Exploring the Benefits of Visual Prompting in Differential Privacy (ICCV 2023)
> >
> >[11] Tang et al. Neural Clamping: Joint Input Perturbation and Temperature Scaling for Neural Network Calibration

---

> ### Author Response · Authors · 2023-11-22
> **Follow up to the Reviewer**
>
> Dear Reviewer,
>
> As the discussion period is closing soon, we'd like to follow up and see if the Reviewer has had a chance to consider our response. We hope the Reviewer can raise the score if the concerns are addressed.
>
> Yours Sincerely,
>
> Authors

---

### Official Review · Reviewer_uT9R · 2023-10-31

**Soundness:** 2 fair
**Presentation:** 2 fair
**Contribution:** 2 fair
**Rating:** 5
**Confidence:** 4

**Summary:**

In this study, the authors introduce NeuralFuse, a data preprocessing module designed to enhance resilience against bit errors arising from low-voltage SRAM, while also offering potential energy savings. Comprehensive tests affirm its efficacy in enhancing models affected by perturbations, ensuring transferability across diverse DNN architectures, and bolstering robustness in weight quantization.

**Strengths:**

1. It focuses on a system aspect of neural network computing: a power-saving method with low voltage operation.
2. The proposed module can work in a plug-and-play manner and does not require retraining the deployed model.

**Weaknesses:**

1. The net benefits of introducing NeuralFuse in tandem with low-voltage operation remain uncertain. While there are energy savings associated with SRAM accesses, these reports overlook the comprehensive energy consumption of NeuralFuse, particularly the MAC operations.
2. Even though there's a notable enhancement in recovered accuracy, it might fall short when juxtaposed with the original accuracy, notably in the case of ResNet50.
3. The significant fluctuation in accuracy suggests that the optimized model may lack consistent predictability.

**Questions:**

1. How does the energy consumption from SRAM accesses compare to the total inference cost of a DNN? While acknowledging that the overall energy consumption hinges on a myriad of factors, providing a general perspective would be insightful.

2. The true efficacy of power savings from the low-voltage operation remains ambiguous. While Table 2 highlights energy savings, it narrowly focuses on the consumption related to SRAM accesses. Given that the large configurations of NeuralFuse exhibit similar MACs to the base models (as seen in Table 7), the feasibility of NeuralFuse, when accounting for its total overhead, merits reconsideration.

3. The unpredictability of model performance under low voltage operation, especially with bit flips at the MSBs, poses challenges for practical implementation. Could you shed more light on its real-world applicability or potential use-cases?

---

> ### Author Response · Authors · 2023-11-21
> **Response to Reviewer uT9R**
>
> Dear Reviewer uT9R,
>
> Thank you for taking the time to review our paper. We are happy to hear that you have found our proposed power-saving method has the advantage of zero-model-parameter updates and can be deployed in a plug-and-play fashion. We have updated our paper based on the reviewers' comments. The following are our responses to your comments on the shortcomings of our paper.
>
> **[W1, Q2]** The net benefits of introducing NeuralFuse in tandem with low-voltage operation remain uncertain…  The true efficacy of power savings from the low-voltage operation remains ambiguous…
>
> **[Response]**  We understand the reviewers' concerns, therefore, we provide the approximation of the end-to-end energy-saving results based on MACs in [Global Response](https://openreview.net/forum?id=Qvoe4wXWFi&noteId=X3C19r9SKy), which can provide more insights into total energy saving under low-voltage scenarios. We observe that only VGG11 with two larger NeuralFuse (ConvL and DeConvL) may increase the total energy. On the contrary, others’ pairs still can save a large amount of energy, which is consistent with the results reported in Table 2.
>
> ---
>
> **[W2]** Even though there's a notable enhancement in recovered accuracy, it might fall short when juxtaposed with the original accuracy, notably in the case of ResNet50.
>
> **[Response]** Although in our cases, ResNet50 may be one of the challenging models to recover the accuracy. However, in Tables 10 and 11, the experimental results show that NeuralFuse can still recover the accuracy of up to 87% (DeConvL) on CIFAR-10 and 93% (UNetL) on GTSRB. These results show that by carefully specifying the NeuralFuse generator, we can still obtain a well-performed NeuralFuse to save the overall energy consumption.
>
> ---
>
> **[W3]** The significant fluctuation in accuracy suggests that the optimized model may lack consistent predictability.
>
> **[Response]** We note that random fluctuation is caused by the random bit flipping caused by low-voltage, and hence larger models, which have more parameters, will suffer more accuracy fluctuation from it. Therefore, the fluctuation is the nature of this challenging task we are addressing (recovering the accuracy against random bit errors), instead of a weakness of our solution.
>
> ---
>
> **[Q1]** How does the energy consumption from SRAM accesses compare to the total inference cost of a DNN?
>
> **[Response]** For a general perspective about the energy consumption from SRAM accesses compared to the total inference cost of a DNN, in Figure 12 in the paper [1], the authors show the total energy consumption in the whole system. We can observe that the energy consumption in the SRAM part (both Buffer and Array) consumes a large amount of total system energy consumption. This is why we emphasize energy saving in SRAM in our paper.
>
> ---
>
> **[Q3]** Could you shed more light on its real-world applicability or potential use cases?
>
> **[Response]** This is really a well-thought-out comment! Previous works have pointed out some possible scenarios that suffer from energy concerns and hence need some strategies to reduce energy consumption. For example, in [2, 3], the authors mention that due to the high computation cost of CNN processing and some DNN-based vision algorithms, they will incur high energy consumption. This will significantly reduce the battery life of battery-powered devices, indirectly impacting the user experience of the devices. Therefore, to avoid the aforementioned issues, we can mitigate the device's energy consumption by lowering the operating voltage and then incorporating NeuralFuse to recover model performance, reducing the side effects caused by low voltage.
>
> > [1] Chen et al. Eyeriss: A Spatial Architecture for Energy-Efficient Dataflow for Convolutional Neural Networks (ISCA 2016)
> >
> > [2] Yang et al. Designing energy-efficient convolutional neural networks using energy-aware pruning (CVPR 2017)
> >
> > [3] Yang et al. Ecc: Platform-independent energy-constrained deep neural network compression via a bilinear regression model (CVPR 2019)

---

> > ### Comment · Reviewer_uT9R · 2023-11-22
> > **Responses**
> >
> > Thank you for your response. However, it seems that the primary concern about energy efficiency has not been directly addressed. The power consumption increase introduced by NeuralFuse is not solely due to the network's memory access. Although the updated formula focuses on the number of operations of NeuralFuse and the Base model, it is primarily the cost of MAC operations at the same scale as the original network that raises the concern. Therefore, the method's claimed power advantage remains a point of discussion. I will keep my score considering this concern.

---

> ### Author Response · Authors · 2023-11-22
> **Response to Primary Concern**
>
> We thank the reviewer for the prompt response. However, we respectfully disagree that the suggested point is a weakness of our method. First, we note that in our experiments, one can *always* utilize a smaller-scale NeuralFuse to recover the accuracy and save the *overall energy consumption* to some degree. For example, although using ConvL or DeConvL along with base model VGG11 for CIFAR-10 implies an increase in energy consumption, using other *smaller-scale* generators (as listed below), we can still save the overall energy and recover the base model’s accuracy.
>
> - ConvS: save 23.9% energy, and recover 24.1​​% accuracy to 66.3%
> - DeConvS: save 16.0% energy, and recover 26.0% accuracy to 68.2%
> - UNetL: save 3.6% energy, and recover 41.4% accuracy to 83.6%
> - UNetS: save 23.7% energy, and recover 30.5% accuracy to 72.7%
>
> We also note that the model size of UNetS or ConvS is about 15 times smaller than that of VGG11, meaning that they are not at the same scale.
>
> [The MACs-Based energy saving percentage (%) on VGG11 (full results are  in Table 8)]
> |       | ConvL | ConvS | DeConvL | DeConvS | UNetL   | UNetS    |
> |-------|-------|-------|---------|---------|---------|----------|
> | VGG11 | -21.8 | **23.9**  | -11.5   | **16.0**    | **3.6** | **23.7** |
>
> [MAC values of all base models and generators (full results are in Table 7)]
> |     |ResNet18| ResNet50 | VGG11 | VGG16   | VGG19    |
> |---|------|------|-----|-----|-----|
> | MACs| 557.14M| 1.31G  | **153.5M**  | 314.43M | 399.47M  |
>
>
> |     | ConvL  | ConvS  | DeConvL | DeConvS | UNetL    | UNetS   |
> |---|------|------|-----|-----|-----|-----|
> | MACs| 80.5M  | **10.34M** | 64.69M  | **22.44M**  |**41.41M**|**10.58M** |
>
> We would also like to point out that when reporting the overall end-to-end MAC-based energy consumption in the rebuttal, as clearly specified in [Golbal Response](https://openreview.net/forum?id=Qvoe4wXWFi&noteId=X3C19r9SKy), we compare the energy saving to the original base model in nominal voltage mode. Therefore, a positive energy-saving ratio means that the overall energy consumption, when measured by the end-to-end MAC operations, can be reduced. Moreover, as mentioned in [2], MAC operations are indeed proportional to the actual energy consumption.
>
> We hope this explanation can address the primary concern. We are at the reviewer’s disposal to answer any further questions.
>
> > [2] Yang et al. Designing energy-efficient convolutional neural networks using energy-aware pruning (CVPR 2017)

---

### Official Review · Reviewer_YHTK · 2023-10-31

**Soundness:** 2 fair
**Presentation:** 4 excellent
**Contribution:** 2 fair
**Rating:** 5
**Confidence:** 5

**Summary:**

This paper aims to tackle the accuracy drop introduced by the increasing bit error rate under the low-voltage scheme by finding a more robust input representation. An error-resistant input transformation is proposed by utilizing a trainable generator, and a modified training loss is utilized to optimize the predicted outputs with/without bit-error injection. The experiments show an obvious accuracy improvement compared to the baseline.

**Strengths:**

* Neat paper structure and easy-to-follow content.
* A simple add-on strategy that can be used in access-limited scenarios.
* Extensive analysis of different generator architectures.

**Weaknesses:**

* Lack of discussion of introduced overhead of the generator modules. For ImageNet-10, the best generator architecture, UNet-L, has 2.03G MACs. The introduced extra computation cost is significant compared to the vanilla model (ResNet18 only has 1.82 G MACs). It raises the concern that the introduced overhead for the generator is too large compared to the classifier, making the proposed strategy unrealistic. The author only discusses the energy of SRAM access without considering the computation energy and latency.
* The introduced generator modules may dilute the energy efficiency brought by the low-voltage scheme. Based on Appendix E, the total computations are very large. A more ideal accuracy-saving method should introduce less overhead.
* Lack of comparison with other error-resistant methods for bit-error rate. The author should add a comparison with other methods to show whether the costly input transformation is worth.

**Questions:**

* Could the author provide a more complete overhead analysis of the introduced generator? The author should show the introduced energy cost of computation (both memory and computation) and extra latency overhead in 4.4. The paper would be more meaningful if it saved accuracy under small overhead.
* Could the author compare with other error-mitigation methods for bit error in SRAM?

---

> ### Author Response · Authors · 2023-11-21
> **Response to Reviewer YHTK**
>
> Dear Reviewer YHTK,
>
> Thank you for your effort in reviewing our paper, and we appreciate your insightful comments and suggestions! We have updated our paper based on the reviewers' comments. To your questions, we address our responses in the following.
>
> **[W1, Q1]** Lack of discussion of introduced energy overhead and extra latency of the generator modules.
>
> **[Response]** We understand the reviewers' concerns; we acknowledge that when evaluating the ImageNet-10 dataset, utilizing UNetL with base model ResNet18, the resulting energy consumption would be higher than the vanilla model. However, we would like to emphasize that this is a trade-off, and it should not be considered unrealistic. When utilizing NeuralFuse to recover the low-voltage-induced accuracy drop, a suitable architecture of the NeuralFuse generator should be considered to reduce the overall energy consumption. Understandably, one utilizing a small base model should not choose such a NeuralFuse generator that is larger than it.  Take ResNet18 for example, we can choose UNetS, which has the MAC value of only 518.47M (less than ResNet18, 1.82G) and can still recover the accuracy up to 86%.
>
> Regarding the computation energy, we did not report the *true* end-to-end energy consumption as it will depend on various factors (as mentioned in Section 4.1). However, MACs (multiply-accumulate operations) report the computation required to perform a forward pass through a neural network [1, 2, 3, 4], and we believe it is a sufficient representation of the energy consumption of the computation. Therefore, we provide the approximation of the end-to-end energy consumption by using MACs in [Global Response](https://openreview.net/forum?id=Qvoe4wXWFi&noteId=X3C19r9SKy).
>
> In the table below, we report the latency (batch_size=1, CIFAR-10/ImageNet-10 testing dataset) of utilizing the different NeuralFuse generators with two different base models, ResNet18 and VGG19. We can see that although NeuralFuse indeed brings a certain degree of extra latency, we argue that this is an unavoidable factor; however, since the latency is measured on a general-purpose GPU (i.e. V100), when the base model and NeuralFuse are deployed on a custom accelerator, we believe this delay will be further reduced.
>
> [NeuralFuse Extra Latency Compared to Only Base Model]
> | | ResNet18 (CIFAR-10) | VGG19 (CIFAR-10) | ResNet18 (ImageNet-10) | VGG19 (ImageNet-10) |
> |-:|-:|-:|-:|-:|
> | Base Model|5.84 ms|5.32 ms|6.21 ms|14.34 ms|
> |+ ConvL | 9.37 ms (+3.53)|8.96 ms (+3.64)|10.51 ms (+4.30)|17.66 ms (+3.32)|
> |+ ConvS | 7.86 ms (+2.02)|7.40 ms (+2.08)|8.28 ms (+2.07)|16.72 ms (+2.38)|
> |+ DeConvL | 9.18 ms (+3.34)|8.59 ms (+3.27)|10.07 ms (+3.86)|17.24 ms (+2.90)|
> |+ DeConvS | 7.49 ms (+1.65)|7.04 ms (+1.72)|7.79 ms (+1.58)|15.67 ms (+1.33)|
> |+ UNetL | 10.69 ms (+4.85)|10.06 ms (+4.74)|11.14 ms (+4.93)|18.54 ms (+4.20)|
> |+ UNetS | 10.63 ms (+4.79)|10.13 ms (+4.81)|11.36 ms (+5.15)|18.60 ms (+4.26)|
>
> ---
> **[W2]** The introduced generator modules may dilute the energy efficiency brought by the low-voltage scheme.
>
> **[Response]**  We understand the reviewer’s concern. Actually, in Table 2, we have already considered both the base model and NeuralFuse energy consumption to calculate the energy saving ratio. Besides, we also show the energy saving results based on MACs in [Global Response](https://openreview.net/forum?id=Qvoe4wXWFi&noteId=X3C19r9SKy), which consider the overall energy consumption. The results show that only VGG11 with two larger NeuralFuse (ConvL and DeConvL) may increase the total energy. On the contrary, others’ pairs still can save a large amount of energy, consistent with our original findings. Although our approach dilutes the amount of energy saving achieved from low voltages, given that our approach is model-agnostic and more broadly applicable to a variety of scenarios, especially models with limited access. This also demonstrates the versatility of NeuralFuse, including adaptability, portability, transferability, and generality.
>
> ---
>
> **[W3, Q2]** Lack of comparison with other error-resistant (or error-mitigation) methods for bit-error rate.
>
> **[Response]** We have provided the experimental results based on a robust model trained with adversarial weight perturbation in Appendix M. Our results show that although the weights are perturbed during training, the trained robust models still can not resist the random bit flips (The perturbed accuracy of PerAct ResNet18 is only about 23%). The results also show that using input transformation based on NeuralFuse is indeed for recovering the accuracy.
>
> > [1] Henderson et al. Towards the Systematic Reporting of the Energy and Carbon Footprints of Machine Learning (JMLR 2020)
> >
> > [2] Howard et al. MobileNets: Efficient Convolutional Neural Networks for Mobile Vision Applications (2017)
> >
> > [3] Sandler et al. MobileNetV2: Inverted Residuals and Linear Bottlenecks (CVPR 2018)
> >
> > [4] Schwartz et al. Green AI Commun. (ACM 2020)

---

> > ### Comment · Reviewer_YHTK · 2023-11-22
> >
> > I have read the response. The proposed methods introduce significant latency overhead, e.g., the most efficient DeConvS on ResNet18 cifar10 adds an extra 28% latency. I disagreed that the latency overhead would be reduced on a custom accelerator, as it typically has fewer computation units.
> > I would like to keep my current score as the obvious overhead which cannot be overlooked.

---

> > > ### Author Response · Authors · 2023-11-22
> > >
> > > We thank the reviewer for the feedback. We understand that the reviewer is disappointed about the additional latency overhead, which we believe is an inevitable tradeoff for reducing energy consumption in our setting. Although latency is not the major focus of this paper, we envision that one can design an even more lightweight version of NeuralFuse module or use model compression techniques on the NeuralFuse model to reduce the latency.

---

### Official Review · Reviewer_Qmac · 2023-10-31

**Soundness:** 3 good
**Presentation:** 4 excellent
**Contribution:** 4 excellent
**Rating:** 8
**Confidence:** 3

**Summary:**

This work presents an add-on module that can be added to image classifiers when they are employed/inferenced in a low-power and error prone accelerator. The module is trained by various perturbated models (models that run on machines with bit errors in SRAMs). The proposed module can be trained on two real-life scenarios: 1) relaxed access and 2) restricted access. The extensive experimental results show that the proposed method is effective in error resiliency and power saving.

**Strengths:**

The paper presents an novel idea of adding a module to any image classifiers where the image model can suffer from low-voltage induced errors. This approach does not require retraining of the models and can be applied to any proprietary-protected DL models.

The extensive experiments show the effectiveness of the work. The paper is well written and organized.

As large models are being developed and deployed around the world, the proposed method can save significant energy and pave the way to greener AI. Although the work is only focused on the image classifier, it opens a door to robust DL in other domains.

**Weaknesses:**

The work assumes that the NeuralFuse generator can be employed on the hardware of no-error voltage. To justify this claim, it would be great if there is a comparison of the sizes (number of parameters) between NeuralFuse generator and the classifier.

**Questions:**

The review can see the architectures of the generators in the appendix. How are the detailed architecture of generators decided? Any insights on the architecture of the NeuralFuse generator?

---

> ### Author Response · Authors · 2023-11-21
> **Response to Reviewer Qmac**
>
> Dear Reviewer Qmac,
>
> Thank you for your encouraging review, and for recognizing our work is novel and well-written, and that extensive experiments are performed to demonstrate the effectiveness of the proposed method. We are thrilled that you enjoyed our paper. We have updated our paper based on the reviewers' comments. To your questions, we address our responses in the following.
>
> **[W1]** A comparison of the sizes (number of parameters) between NeuralFuse generator and the classifier.
>
> **[Response]** Thank you for the constructive feedback. We have provided the size comparison of the NeuralFuse and the classifier in Table 7 in Appendix E, which includes the parameters and multiply-accumulate operations (MACs) of all the base models and the NeuralFuse generators, in the same units as Bejnordi et al. [1] used. In general, our proposed NeuralFuse generators are smaller than classifiers, either on total model parameters or MAC values.
>
> ---
>
> **[Q1]** How are the detailed architecture of generators decided? Any insights on the architecture of the NeuralFuse generator?
>
> **[Response]** Thank you for raising this fundamental question. When it comes to deciding which architecture should be adopted to address this particular issue, we have two main goals: 1) efficiency (so the overall energy overhead is decreased) and 2) robustness (so that it can generate robust patterns on the input image and overcome the random bit flipping in subsequent models). As mentioned in Section 4.1, the design of ConvL is inspired by Nguyen and Tran [2], in which the authors utilize a similar architecture to design an input-aware trigger generator, and have demonstrated its efficiency and effectiveness. Furthermore, since ConvL is an encoder-decoder-based model, we attempted to enhance it by replacing the *Upsampling* layer with a *Deconvolution* layer, leading to the creation of DeConvL. The UNetL-based NeuralFuse draws inspiration from Ronneberger et al. [3], known for its robust performance in image segmentation, and thus, we incorporated it as one of our architectures. Lastly, ConvS, DeConvS, and UNetS are *scaled-down* versions of the model designed to reduce computational costs and total parameters. These architectural variations also demonstrate the trade-off between energy saving and the accuracy recovering rate, and a practitioner can specify a particular architecture based on the need.
>
>
> > [1] Bejnordi et al. Batch-shaping for learning conditional channel gated networks (ICLR 2021)
> >
> > [2] Nguyen and Tran. Input-aware dynamic backdoor attack (NeurIPS 2020)
> >
> > [3] Ronneberger et al. U-net: Convolutional networks for biomedical image segmentation (MICCAI 2015)

---

> > ### Comment · Reviewer_Qmac · 2023-11-22
> >
> > Acknowledge of the responses.
> > It would be good to include these points into the paper. In particular, the trade-off reasoning will assist future work and industry adoption.

---

> ### Author Response · Authors · 2023-11-22
>
> We thank the reviewer for the prompt response. We totally agree that "the trade-off reasoning will assist future work and industry adoption", and we will include these discussions in the future version.

---

### Author Response · Authors · 2023-11-21
**General Response**

We are very grateful to Reviewers Qmac, YHTK, uT9R, 7gLE for your efforts in reviewing our paper and for your many insightful questions and helpful suggestions. With your assistance, we are committed to enhancing the clarity and conciseness of our paper. We have provided answers to your specific questions and concerns, and have incorporated our response in the paper revision (highlighted in $\color{blue} \text{blue}$).

We hope the revised version has addressed the reviewers’ questions and concerns. We are just one post away from answering any follow-up questions the reviewers may have, and we look forward to the reviewers’ feedback.

---
Due to some questions from YHTK, uT9R  being similar, we would like to use this space to respond to them.

**[YHTK (W1, Q1), uT9R (W1, Q2)]** Comprehensive Energy Consumption

**[Response]** We understand that the reviewers would like to learn more about the net benefits and the true efficacy of power savings, instead of merely consideration of SRAM access. In fact, the energy consumption of computation and memory accesses are both proportional to MACs, which means we could estimate the overall energy consumption via MACs [1]. Therefore, we can then use the MAC values to approximate the end-to-end energy consumption of the whole model. Assume that all values are stored on SRAM and that a MAC represents single memory access. The corresponding MACs-based energy saving percentage (MAC-ES, \%) can be derived from the Equation below (c.f. Section 4.4), and results can be found in the following Table.

With this new analysis, we can observe that only VGG11 with two larger NeuralFuse (ConvL and DeConvL) may increase the total energy. On the contrary, others’ model pairs also can save a large amount of energy, which is consistent with the results reported in Table 2.


$\text{MAC-ES} = \frac{\text{MACs}\_{\text{base model}} \cdot \text{Energy}\_{\text{nominal voltage}}  - \big(\text{MACs}\_{\text{base model}} \cdot \text{Energy}\_{\text{low-voltage-regime}} + \text{MACs}\_{\text{NeuralFuse}} \cdot \text{Energy}\_{\text{NeuralFuse at nominal voltage}}\big)}{\text{MACs}\_{\text{base model}} \cdot \text{Energy}\_{\text{nominal voltage}}}\times 100\\%$

|          | ConvL | ConvS | DeConvL | DeConvS | UNetL | UNetS |
|----------|-------|-------|---------|---------|-------|-------|
| ResNet18 |  16.2 |  28.7 |    19.0 |    26.6 |  23.2 |  28.7 |
| ResNet50 |  24.5 |  29.8 |    25.7 |    28.9 |  27.4 |  29.8 |
| VGG11    | -21.8 |  23.9 |   -11.5 |      16 |   3.6 |  23.7 |
| VGG16    |     5 |  27.3 |      10 |    23.5 |  17.4 |  27.2 |
| VGG19    |  10.4 |    28 |    14.4 |      25 |  20.2 |    28 |

> [1] Yang et al. Designing energy-efficient convolutional neural networks using energy-aware pruning (CVPR 2017)

---

### Meta-Review · Area_Chair_saZt · 2023-12-07

**Metareview:**

Summary:

This paper proposes NeuralFuse, a model-agnostic approach that learns input transformations to generate error-resistant data representations. NeuralFuse dynamically adds a correction term to the model input to protect the DNNs in both nominal and low-voltage scenarios and can be applied to DNNs with limited access. Experimental results show that NeuralFuse can reduce SRAM memory access energy by up to 20-30% while recovering accuracy by up to 57% at a 1% bit error rate.

Strengths:

- The paper presents an novel idea of adding a module to any image classifiers where the image model can suffer from low-voltage induced errors.
- This approach does not require retraining of the models and can be applied to any proprietary-protected DL models.
- The extensive experiments show the effectiveness of the work.
- The paper is well written and organized.
- A simple add-on strategy that can be used in access-limited scenarios.
- Extensive analysis of different generator architectures.
- The proposed module can work in a plug-and-play manner and does not require retraining the deployed model.
- The idea is quite interesting. Without error-aware training (adversarial training), it learns input-dependent, model-agnostic calibrator for the model input, the DNN’s accuracy can be protected.
- The proposed neural network can protect the DNN accuracy while still showing energy efficiency benefits.
- It thoroughly investigates the transferability to different error rates, model architecture, and quantization bitwidth.

Weaknesses:

- Lack of discussion of introduced overhead of the generator modules.
- Concern that the introduced overhead for the generator is too large compared to the classifier, making the proposed strategy unrealistic.
- The authors only discuss the energy of SRAM access without considering the computation energy and latency.
- The introduced generator modules may dilute the energy efficiency brought by the low-voltage scheme.
- Based on Appendix E, the total computations are very large. A more ideal accuracy-saving method should introduce less overhead.
- Lack of comparison with other error-resistant methods for bit-error rate. The author should add a comparison with other methods to show whether the costly input transformation is worth.
- The net benefits of introducing NeuralFuse in tandem with low-voltage operation remain uncertain.
- While there are energy savings associated with SRAM accesses, these reports overlook the comprehensive energy consumption of NeuralFuse, particularly the MAC operations.
- Even though there's a notable enhancement in recovered accuracy, it might fall short when juxtaposed with the original accuracy, notably in the case of ResNet50.
- The significant fluctuation in accuracy suggests that the optimized model may lack consistent predictability.
- The transferability on different error rate and model size is not very good, according to Table 1, which means re-training is still required for different model/dataset/SRAM voltages.
- The energy saving is only ~20% by reducing SRAM voltage, while the accuracy drop is beyond 1%. It needs some justification on this trade-off.
- The method seems to be equivalent to adding extra layers in the early stage of the network and train it with noise-aware training. Why not train other layers with the memory error?
- It is not very intuitive that weight errors in all layers (even MSB flips) can be well protected by only changing the model input. More explanation is needed to justify this.

Recommendation:

This is a borderline paper. A majority of reviewers weakly recommend rejection. I, therefore, recommend rejecting the paper and encourage the authors to use the feedback provided to improve the paper and resubmit to another venue.

**Justification For Why Not Higher Score:**

A majority of reviewers lean towards rejection. The paper as many significant limitations as stated by the reviewers:

- Lack of discussion of introduced overhead of the generator modules.
- Concern that the introduced overhead for the generator is too large compared to the classifier, making the proposed strategy unrealistic.
- The authors only discuss the energy of SRAM access without considering the computation energy and latency.
- The introduced generator modules may dilute the energy efficiency brought by the low-voltage scheme.
- Based on Appendix E, the total computations are very large. A more ideal accuracy-saving method should introduce less overhead.
- Lack of comparison with other error-resistant methods for bit-error rate. The author should add a comparison with other methods to show whether the costly input transformation is worth.
- The net benefits of introducing NeuralFuse in tandem with low-voltage operation remain uncertain.
- While there are energy savings associated with SRAM accesses, these reports overlook the comprehensive energy consumption of NeuralFuse, particularly the MAC operations.
- Even though there's a notable enhancement in recovered accuracy, it might fall short when juxtaposed with the original accuracy, notably in the case of ResNet50.
- The significant fluctuation in accuracy suggests that the optimized model may lack consistent predictability.
- The transferability on different error rate and model size is not very good, according to Table 1, which means re-training is still required for different model/dataset/SRAM voltages.
- The energy saving is only ~20% by reducing SRAM voltage, while the accuracy drop is beyond 1%. It needs some justification on this trade-off.
- The method seems to be equivalent to adding extra layers in the early stage of the network and train it with noise-aware training. Why not train other layers with the memory error?
- It is not very intuitive that weight errors in all layers (even MSB flips) can be well protected by only changing the model input. More explanation is needed to justify this.

**Justification For Why Not Lower Score:**

N/A

---

### Decision · Program_Chairs · 2024-01-16

Reject